# Evaluation of a wind tunnel designed to investigate the response of evaporation to changes in the incoming longwave radiation at a water surface

Michael L. Roderick[1], Chathuranga Jayarathne[1], Angus J. Rummery[1], Callum J. Shakespeare[1,2]

[1]Research School of Earth Sciences, Australian National University, Canberra, Australia, 2601.
[2]ARC Centre of Excellence for Climate Extremes, Australian National University, Canberra, Australia, 2601.

*Correspondence to*: Callum J. Shakespeare (callum.shakespeare@anu.edu.au)

**Abstract.** To investigate the sensitivity of evaporation to changing longwave radiation we developed a new experimental

facility that locates a shallow water bath at the base of an insulated wind tunnel with evaporation measured using an accurate digital balance. The new facility has the unique ability to impose variations in the incoming longwave radiation at the water surface whilst holding the air temperature, humidity and wind speed in the wind tunnel at fixed values. The underlying scientific aim is to isolate the effect of a change in the incoming longwave radiation on both evaporation and surface temperature. In this paper we describe the configuration and operation of the system and outline the experimental design and

approach. We then evaluate the radiative and thermodynamic properties of the new system and show that the shallow water bath naturally adopts a steady state temperature that closely approximates the thermodynamic wet bulb temperature. We demonstrate that the longwave radiation and evaporation are measured with sufficient precision to support the scientific aims.

## 1 Introduction

The Earth's climate system is in some sense like a giant heat engine with water evaporating at the relatively warm surface and

condensing at a relatively colder altitude in the atmosphere. With water the dominant surface cover on the planet, the water cycle emerges as a central component of both the thermodynamics and dynamics of the climate system (Peixoto and Oort, 1992; Pierrehumbert, 2010). Traditionally, the evaporation of water at the surface has been described using bulk-formulae with the evaporation held to depend on the difference in specific humidity between the (saturated) surface and (sub-saturated) atmosphere, the wind speed and a transfer coefficient (WMO, 1977; Monteith and Unsworth, 2008). The use of bulk-formulae

requires measurement of the surface temperature to specify the specific humidity at the (near-saturated) surface. On that approach, it is straight-forward, in principle at least, to conduct experiments using a controlled wind tunnel to measure the evaporation from a water body as a function of surface temperature, specific humidity in the adjacent air and the wind speed. It is also possible to use comprehensive field measurements to derive bulk-formulae for evaporation (Penman, 1948; Thom et

al., 1981; Lim et al., 2012). The same approach can be used to derive bulk-formulae for sensible heat transfer with the gradient given by the difference in temperature between the water surface and overlying air (WMO, 1977).

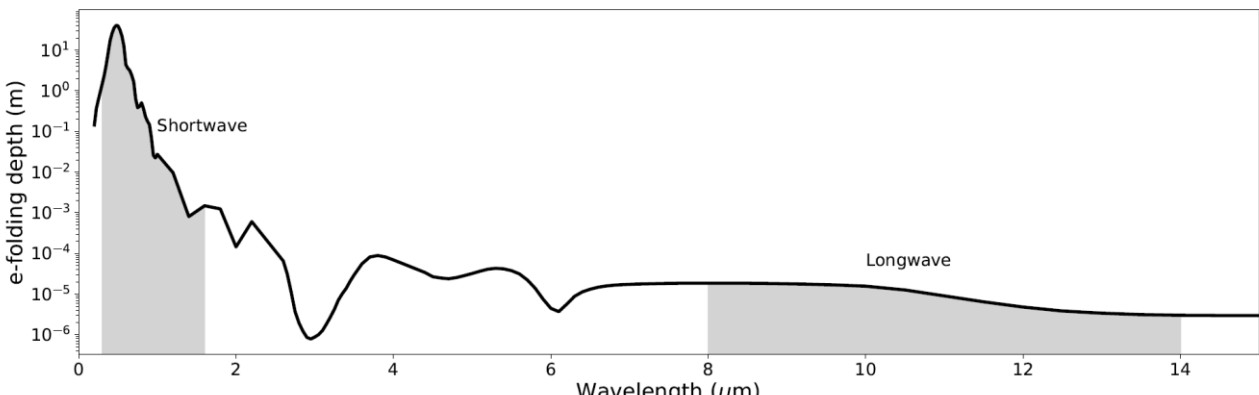

**Figure 1: Characteristic penetration depth of radiation into liquid water at different wavelengths (Irvine and Pollack, 1968; Hale and Querry, 1973). Shaded regions highlight the shortwave (here taken as 0.3-1.6 µm) and longwave (here taken as 8-14 µm) regions of the electromagnetic spectrum. Note the log-scale (y-axis).**

In more detail, the latent heat flux ($LE$) with $L$ the latent heat of vapourisation and $E$ the evaporation rate is typically given by a Dalton-like bulk-formulae (Dalton, 1802) of the form,

$$LE \propto U(q_S(T_S) - q_A) \qquad , \qquad \qquad (1)$$

with $LE$ having a direct dependence on wind speed ($U$) and the difference in specific humidity between the (near-saturated) surface at temperature $T_S$ ($q_S(T_S)$) and the ambient air ($q_A$). The bulk-formulae approach is ubiquitous in heat transfer studies (e.g., see Chapter 6 in Incropera et al, 2017). Once the evaporation has been calculated using the bulk-formulae, in climate science it is standard practice to then construct a comprehensive energy budget for a water surface (e.g., ocean, lake, etc.) by combining the above-noted latent heat flux with the sensible heat flux, incoming and outgoing shortwave and longwave radiative fluxes and also by accounting for energy storage in the water body. Importantly, clear liquid water is relatively transparent to shortwave radiation with a characteristic e-folding absorption depth (i.e., depth at which $1/e$ (~ 37%) of the incident radiation remains) of order 40 m at a (shortwave) wavelength of 0.5 µm (Fig. 1). In contrast, longwave radiation has a characteristic e-folding absorption depth of only $16 \times 10^{-6}$ m at a (longwave) wavelength of 10 µm that is *6 orders of magnitude* smaller than for shortwave radiation (Fig. 1). It follows that most of the emitted longwave radiation must also emanate from the same depth. With liquid (and solid) water having an emissivity (and hence longwave absorption) close to unity, we anticipate that longwave radiation must impact the near-surface energy balance on almost instantaneous time scales. To give a useful numerical example, an e-folding absorption depth of only 16 µm implies that 95% ($= 1 - e^{-48/16}$) of the incoming longwave radiation will have been absorbed after travelling 48 µm below the ocean surface. Hence, for the example

calculation we assume that the global annual average incoming longwave radiation at the surface of ~ 342 W m$^{-2}$ (Wild et al.,

2013) was completely absorbed in the top 50 µm of the ocean. Without any other heat transfer, this 50 µm layer of water would

warm by around 2°C every second. The fact that this warming rate is not observed – even in a perfectly still ocean without

surface overturning -- implies a very efficient means of shedding that heat (by latent and sensible heat and by outgoing

longwave radiation) into the atmosphere  and/or by conductive/convective fluxes into the interior of the ocean (Monteith and

Unsworth, 2008; Peixoto and Oort, 1992;  Saunders, 1967; Woolf et al., 2016).


As noted previously, the bulk-formulae for evaporation in widespread use specify evaporation in terms of the difference

between specific humidity at the surface and in the adjoining air and the wind speed (Eqn 1), with no explicit reference to the

radiative fluxes. We anticipate that such bulk formulae for evaporation from a water body are reasonable when the incoming

and outgoing longwave radiative fluxes are equal because their effects would cancel. However, under the more common

oceanic conditions, the incoming and outgoing longwave radiative fluxes do not cancel and may be important for evaporation

because those longwave fluxes would lead to a near-immediate response since they occur only a small (10-20 µm) distance

from the evaporating surface. If the longwave fluxes were important for evaporation as we have inferred, but did not cancel,

then the Dalton-type formulae in widespread use (e.g. Eqn 1) would not be a valid description of the evaporation process.

Previous theoretical and laboratory-based research has reported that any difference between incoming and outgoing longwave

radiative fluxes may need to be considered an important part of the evaporative bulk-formulae (Nunez and Sparrow, 1988;

Sparrow and Nunez, 1988) thereby invalidating the Dalton-type formulae. The implication here is that the formulation of the

widely used bulk-formulae (Eqn 1) to calculate evaporation (and by inference also for sensible heat) may need to be re-

considered to directly include the potentially important direct effect of longwave radiation on evaporation. Besides the above-

noted Nunez-Sparrow study, we are not aware of any other experimental work on this topic.


To support an investigation of the bulk formulae for evaporation we sought to develop a new experimental system that could

measure and/or control the traditional variables considered in mass transfer studies of evaporation (see Eqn 1, $U$, $q_S(T_S)$, $q_A$).

The innovative feature of the new system is the ability to independently vary the incoming longwave radiation at the water

surface whilst holding the other variables fixed. The scientific rationale of this approach was to isolate the effect of a change

in the incoming longwave radiation on both evaporation and surface temperature. To our knowledge this experimental

approach has not previously been attempted and we found that it presented numerous experimental challenges. In this paper

we describe the experimental wind tunnel and present our evaluation of the overall radiative and thermodynamic behaviour of

the system. The paper is set out as follows. In section 2 we describe both the design and operation of the experimental wind

tunnel. In section 3 we describe the measurement of the incoming and outgoing longwave radiation at the water surface. In

section 4, we describe the thermodynamic behaviour of the experimental wind tunnel. Importantly, we show that the steady

state temperature of the shallow water bath closely approximates the theoretical wet bulb temperature. In section 5 we evaluate

the magnitude and uncertainty of the radiative and evaporative fluxes to ascertain whether the system can be used for the intended purpose. In section 6 we present a discussion and conclusions.

## 2 Design and Operation

In this section we describe the configuration (section 2.1), underlying energy balance (section 2.2) and practical operation of the wind tunnel (section 2.3) and conclude with a description of the experimental design (section 2.4) followed by a brief summary (section 2.5).

### 2.1 Configuration of the wind tunnel

The wind tunnel layout is shown in Fig. 2. The wind tunnel was constructed of closed cell foam (density of 60 kg m$^{-3}$, cross
section of 300 × 300 mm, 2550 mm total length) located on a laboratory bench with a recirculating flow of air passed through heating duct located under the bench. During experiments the wind speed was controlled using a variable speed fan located in series within the heating duct (see [2] in Fig. 2a) and measured using a hot wire anemometer (Sierra Instruments: Model No. 600, not visible in Fig. 2a but located downstream of the water bath; $U$ in Fig. 2b). The same closed cell foam material was used to construct a shallow water bath (diameter 200 mm, 8 mm depth, see [1] in Fig. 2a, also see Fig. 2b) that sat on a digital
balance. The shallow water bath and the base of the tunnel elsewhere were painted using commercial waterproof paint (longwave emissivity ~ 1, results not shown) to ensure the surface was impermeable to water. The rate of change of the mass of water in the water bath was used to determine the evaporation rate from the shallow water bath ($E$ in Fig. 2b). During routine evaporation experiments, the radiometer (Kipp and Zonen: Model CNR1 net radiometer) was located in the laboratory (in the cardboard box sitting on top of the tunnel in Fig. 2a) and used to directly measure the incoming longwave radiation arriving at
the top of the outer film ($R_{i,F2}$ in Fig. 2b). The facility could also be operated in a radiative calibration mode. For that, the shallow water bath was removed and replaced by the (same) radiometer (Kipp and Zonen: Model CNR1 net radiometer) that was custom mounted onto a closed cell foam base so that the centre of the longwave sensor was at exactly the same horizontal and vertical position as the centre of the water surface in the shallow water bath. The radiative calibration experiments were used to verify and subsequently refine a radiative transfer model used to estimate $R_{i,S}$ (see section 3.6).


(a)

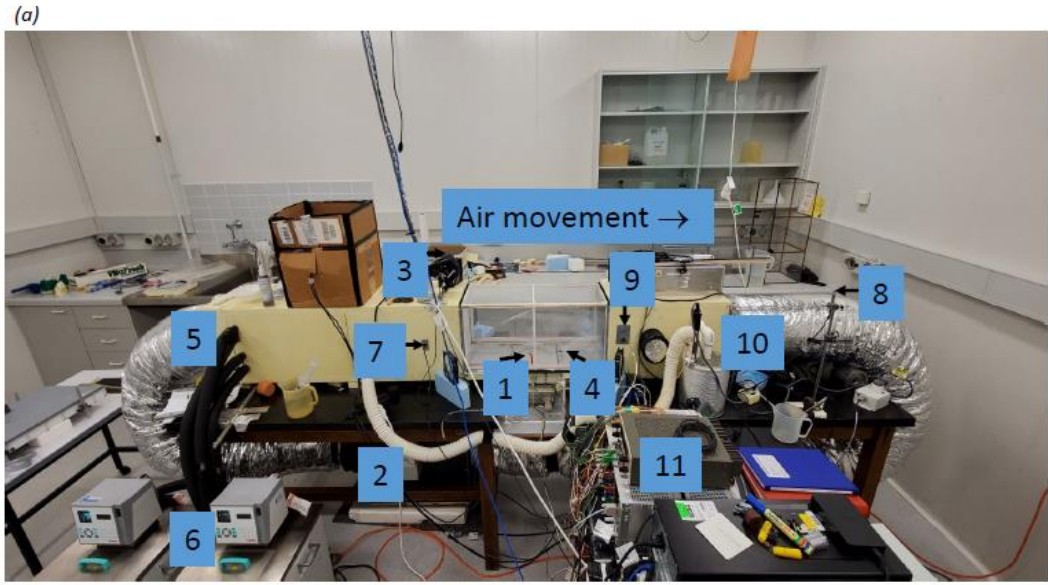

(b)

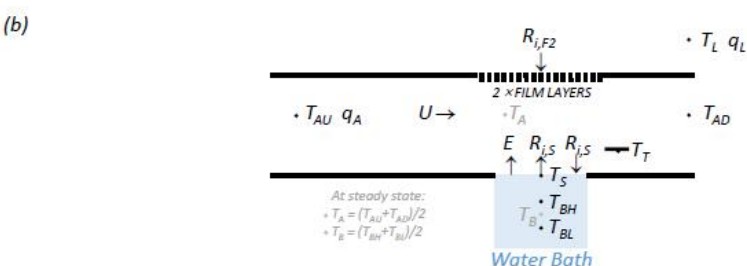

**Figure 2: Configuration of the wind tunnel. (a) Photograph of the wind tunnel in the temperature controlled room of the Geophysical Fluid Dynamics Laboratory. Key numbers as follows: [1] Water bath and digital balance (AND Corporation: Model GX-6100); [2] Variable speed fan located within the tunnel; [3] Thermal camera (FLIR: Model E50); [4] Camera calibration spot (used for thermal camera calibration); [5] Radiator (for air temperature control) located within the tunnel; [6] Constant temperature water bath (Julabo: Model PP50); [7] Humidity/Temperature sensor (for measuring tunnel air, VAISALA: Model HMP140); [8] Humidity/Temperature sensor (for measuring laboratory air, VAISALA: Model HMP140); [9] Temperature sensor (thermistor for measuring tunnel air, Thermometrics NTC: Model FP07DA103N ); [10] Vapour source (humidifier for humidity control of tunnel air); [11] Digital controller. (b) Schematic diagram showing the key thermodynamic and radiative variables (see Table 1). Note that $T_T$ denotes the temperature of the camera calibration spot (see [4] in (a)).**

| Variable | Units | Description |
|---|---|---|
| $R_{i,F2}$ | W m$^{-2}$ | Incoming longwave radiation at the top of the film. Measured by radiometer. |
| $R_{i,S}$ | W m$^{-2}$ | Incoming longwave radiation at the water surface. Calculated using $R_{i,F2}$ measurement. |
| $R_{o,S}$ | W m$^{-2}$ | Outgoing longwave radiation at the water surface. Measured indirectly by thermal camera. |
| $T_S$ | °C | Skin temperature at surface of water body. Measured indirectly by thermal camera. |
| $E$ | kg m$^{-2}$ s$^{-1}$ | Evaporation from water body. Measured by balance. |
| $L$ | J kg$^{-1}$ | Latent heat of vapourisation (~ 2.4 MJ kg$^{-1}$). |
| $LE$ | W m$^{-2}$ | Latent heat flux. Measured by balance ($E$) and converted to heat equivalent (using $L$). |
| $U$ | m s$^{-1}$ | Wind speed in tunnel. Measured by hot wire anemometer. |
| $T_{AU}$ | °C | Temperature of air in tunnel upstream of water body. Measured by T/humidity sensor. |
| $q_A$ | kg kg$^{-1}$ | Specific humidity of air in tunnel upstream of water body. Measured by T/humidity sensor. |
| $T_{AD}$ | °C | Temperature of air in tunnel downstream of water body. Measured by thermistor. |
| $T_L$ | °C | Temperature of air in laboratory. Measured by T/humidity sensor. |
| $q_L$ | kg kg$^{-1}$ | Specific humidity of air in laboratory. Measured by T/humidity sensor. |
| $T_{AD}$ | °C | Temperature of air in tunnel downstream of water body. Measured by thermistor. |
| $T_A$ | °C | Steady state temperature of air in tunnel. Calculated using ($T_{AD}+T_{AU}$)/2. |
| $T_T$ | °C | Temperature of camera calibration spot. Measured by thermocouple. |
| $T_{BH}$ | °C | Temperature of water 5 mm from bottom of (8 mm deep) water bath. Measured by thermocouple. |
| $T_{BL}$ | °C | Temperature of water 1 mm from bottom of (8 mm deep) water bath. Measured by thermocouple. |
| $T_B$ | °C | Steady state temperature of water in (8 mm deep) water bath. Calculated using ($T_{BH}+T_{BL}$)/2. |
| $T_W$ | °C | Theoretical wet bulb temperature. |
| $G$ | W m$^{-2}$ | Rate of change of enthalpy in the water body. Defined using $T_B$. |
| $\sigma$ | W m$^{-2}$ K$^{-4}$ | Stefan-Boltzmann Constant (= 5.67 x 10$^{-8}$). |
| $\varepsilon_S$ | - | Emissivity of water surface (= 0.95). |
| $\alpha, \beta, \tau$ | - | Bulk reflection ($\alpha$), absorption ($\beta$) and transmission ($\tau$) of a single layer of the plastic film. |

**Table 1   List of key variables**

Two thermocouples (Thermocouples Direct: Model KM1(118)0.25x250) were inserted into the (8 mm deep) shallow water bath to measure the bulk (liquid) water temperature. The 'high' sensor was located 5 mm from the bottom ($T_{BH}$ in Fig. 2b) and the 'low' sensor was located 1 mm from the bottom ($T_{BL}$ in Fig. 2b) of the shallow water bath. The design intent was for the base of the shallow water bath to form a 'no heat flux' condition (i.e., an adiabatic lower boundary). By measuring the temperature in the closed cell foam below the shallow water bath using a temperature probe during typical evaporation experiments (results not shown) we concluded that the design intent was achieved because of the excellent insulation properties of the closed cell foam. Directly above the shallow water bath we located a removable PVC frame (730 mm length) covered by two layers of polyethylene (i.e., plastic) film (Fig. 2a) enclosing a 10 mm air gap between them, with each film being 0.022 mm thick. We found (by trial and error) that the use of two layers of film allowed us to avoid condensation of water onto the interior film (see discussion in section 2.3). We placed silica gel desiccant beads in the air gap to further avoid condensation. Above the PVC frame (and outside the film) we located a thermal camera (FLIR: Model E50, see [3] in Fig. 2a) to measure the surface (skin) temperature of water in the water bath during evaporation experiments ($T_S$ in Fig. 2b). This was an indirect measure since it required corrections to account for modifications to the longwave radiation as it passed through the two plastic films and the intervening moist air (see section 3.4).

On the downstream side of the shallow water bath we installed a small circular copper plate (the 'spot', see [4] in Fig. 2a) painted with commercial paint (longwave emissivity ~ 1, results not shown) to assist with calibration of the thermal camera. The copper 'spot' (~ 1 mm thick) was clearly visible in the thermal imagery and we drilled a hole and inserted a thermocouple (Thermocouples Direct: Model KM1(118)1.0x250) into the underside of the copper 'spot' to measure the temperature of the 'spot' and thereby assist with calibration of the thermal camera measurements that were used to measure $T_S$ (and $R_{o,S}$, see Fig. 2b). As described below, the temperature, humidity and wind speed of air within the tunnel could all be held fixed at user-defined levels. By locating the entire wind tunnel facility within a temperature controlled room (length 6700 mm, width 4600 mm, height 3000 mm) within the Geophysical Fluid Dynamics Laboratory we were able to vary the incoming longwave radiation arriving at the top of the plastic film ($R_{i,F2}$ in Fig. 2b) by changing the air temperature ($T_L$ in Fig. 2b) – and thus the temperature of all surfaces – in the room. Note that the incoming longwave radiation at the top of the film ($R_{i,F2}$ in Fig. 2b) is effectively the blackbody radiation emitted by the walls of the temperature controlled room at temperature $T_L$. With most of that longwave radiation ultimately transmitted through the two film layers to the water surface we were able to experimentally change the incoming longwave radiation arriving at the water surface independently of the air temperature, humidity and wind speed within the tunnel.

In more detail, the air temperature in the tunnel was controlled using a commercial radiator installed within the tunnel upstream of the water bath (see [5] in Fig. 2a) and connected via a recirculating flow to an external constant temperature water bath (see [6] in Fig. 2a). As the air stream moved through the constant temperature radiator, heat conduction ensured the air temperature in the tunnel rapidly equilibrated with the radiator temperature. We measured the temperature and humidity of the air stream

after it had passed through the radiator but still upstream of the water bath (see [7] in Fig. 2a; $T_{AU}$ and $q_A$ in Fig. 2a). Following that the air was passed through a block of plastic straws of cross section of $300 \times 300$ mm and length of 150 mm with each individual straw in the block having a diameter of 4 mm. This block, commonly known as a 'laminarizer' (Huang et al., 2017) engineered a near-laminar flow of air (verified using smoke experiments, results not shown) over the shallow water bath. The air temperature was measured downstream of the shallow water bath (see [9] in Fig. 2a; $T_{AD}$ in Fig. 2b). Finally, we measured

the temperature and specific humidity of air in the laboratory (external to the tunnel) (see [8] in Fig. 2a; $T_L$ and $q_L$ in Fig. 2b). The humidity sensors (see [7] and [8] in Fig. 2a; and respectively $q_A$ and $q_L$ in Fig. 2b) measured the relative humidity and this was converted to specific humidity (Huang, 2018) by assuming the moist air to be an ideal gas with the total air pressure set to 1 bar (i.e., the climatological average for Canberra, Australia).

All sensors were connected to a digital sampling system (see [11] in Fig. 2a) that was interfaced to a standard computer with all data sampling and acquisition controlled using the LabVIEW (National Instruments Corporation) software package. The one exception was the thermal camera which was operated independently using instrument-specific software available (by purchase) from the manufacturer. In post-processing, the thermal camera measurements of surface temperature were merged into the experimental database using time stamps embedded within both data streams. During the experiments all data elements

were sampled at 30 Hz and then averaged to successive 10 second time steps within the LabVIEW control software. The same sampling protocol was used for the thermal imagery.

## 2.2 Energy balance for the experiment

With the experiment conducted indoors we were able to ignore the shortwave radiative fluxes. The energy balance for the experiment is defined at the water surface by,

$$G = R_{i,S} - R_{o,S} - LE - H \qquad , \qquad\qquad\qquad\qquad (2)$$

with $R_{i,S}$ and $R_{o,S}$ the measured incoming and outgoing longwave radiative fluxes and $LE$ the measured latent heat flux as per the previous definitions (Fig. 2b, Table 1). $G$ (W m$^{-2}$) is the rate of change of enthalpy in the water body and is directly measured using temperature measurements in the water bath ($T_{BH}$, $T_{BL}$, $T_B$, Fig. 2b, Table 1). Note that at steady state we have $G = 0$. Finally, $H$ (W m$^{-2}$) is the sensible heat flux from the water surface to the air and this flux was not measured. Instead it

can be calculated when necessary via energy balance (Eqn 2) using the other four measured quantities.

## 2.3 Operation of the wind tunnel

During evaporation experiments both the air temperature and wind speed in the tunnel proved relatively easy to control. The most challenging variable to control was the humidity of air within the tunnel. The experiments were designed so that the pre-determined specific humidity of the tunnel air generally exceeded that in the laboratory which required the addition of water

vapour to the tunnel air to arrive at the pre-determined humidity. For that purpose, we used an independently controlled electrical heater element immersed in a water bath to generate warm water vapour that could be vented into the tunnel on

demand (see [10] in Fig. 2a). Occasionally we would overshoot the pre-determined specific humidity of the tunnel air and we used a condenser to remove excess water vapour. For that we installed a temperature-controlled copper plate on the base of the tunnel (not visible but located within the tunnel downstream of [10] in Fig. 2a). The copper plate was connected to another constant temperature water bath (again not visible but of the same type as [6] in Fig. 2a) that recirculated water through a network of channels within the copper plate. By cooling the copper plate as required we were able to engineer a cold surface onto which excess water vapour could be condensed and routed to an external drain on demand.

Typical operations would begin each day by filling the shallow water bath to a pre-determined mass (we used ~ 250 (± 25) g of water and equivalent to ~ 8 mm water depth) and by allowing the externally controlled radiator (see [5] and [6] in Fig. 2a) to come to a steady state temperature. Each of the numerous temperature sensors were then checked against the portable laboratory reference (HART Scientific: Model 1521) and any necessary (minor) offset adjustments made within the LabVIEW control software.

### 2.4 Experimental design

As part of the overall experimental program, we conducted both radiation and evaporation sub-experiments at pre-determined combinations of air temperature and specific humidity in the wind tunnel (Fig. 3). The original aim was to sample a regular grid of temperature (15, 25, 35, 45 °C) and specific humidity (5, 15, 25, 35 g kg$^{-1}$) conditions. This range was selected to span the conditions typical of tropical oceans (near surface air of 31°C, 80% relative humidity ~ 20 g kg$^{-1}$) (Priestley, 1966). The lower bound for the specific humidity range was subsequently increased from 5 to 7 g kg$^{-1}$ to avoid (where possible) circumstances where moisture had to be extracted from air in the tunnel. For the radiation calibration experiments the water bath was replaced by the radiometer that was carefully located in exactly the same position (see Section 2.1). We directly measured the incoming longwave radiation that would have been received at the water surface under the prevailing ($T_A$-$q_A$) conditions. This was repeated successfully for all ten predetermined $T_A$-$q_A$ combinations (Fig. 3) with the windspeed set to 2 m s$^{-1}$. To control the incoming longwave radiation arriving at the top of the film ($R_{i,F2}$ in Fig. 2b) we set the laboratory air temperature on the room controller to be either 19°C which we denoted the 'Ambient' condition or to 31°C which we denoted the 'Forced' condition. A change between the 'Ambient' and 'Forced' condition took several hours to equilibrate within the temperature-controlled room and was usually completed overnight. The difference between the 'Forced' (31°C, black body longwave radiative flux of ~ 485 W m$^{-2}$) and 'Ambient' (19°C, black body longwave radiative flux of ~ 413 W m$^{-2}$) conditions gave an experimentally imposed longwave forcing of around 72 W m$^{-2}$ at the top of the film. By this construction we were able to experimentally measure the longwave radiation arriving at the location of the water bath at the base of the tunnel for the twenty different combinations (i.e., ten $T_A$-$q_A$ combinations under either Ambient or Forced longwave conditions). The radiation calibration experiments were conducted first.

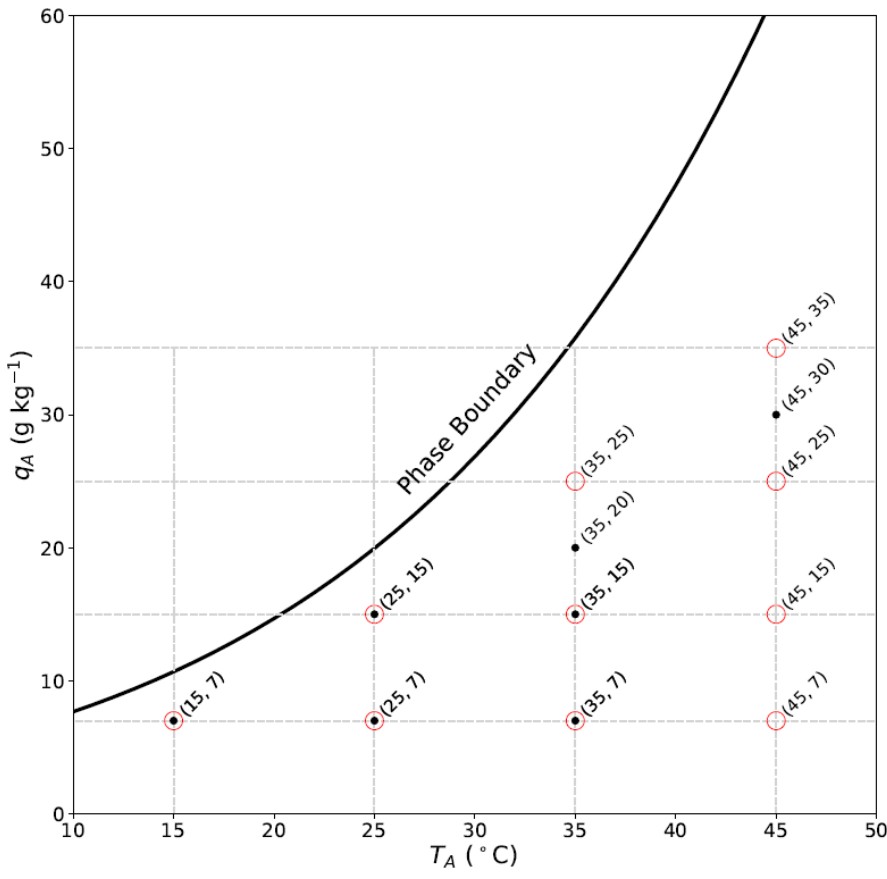

**Figure 3: Layout of the ten radiation sub-experiments (red circles) and seven evaporation sub-experiments (black dots) as a function of air temperature ($T_A$) and specific humidity ($q_A$) inside the wind tunnel. The full line denotes the liquid-vapour phase boundary (i.e., saturation curve, total pressure of 1 bar) computed using an empirical equation (Huang, 2018).**

The basic idea for the evaporation experiments was to follow the same procedure with the addition that at each $T_A$-$q_A$ combination we varied the wind speed over five discrete steps (0.5, 1.0, 2.0, 3.0, 4.0 m s$^{-1}$). Ideally, this would have left us with 100 individual evaporation sub-experiments (the same ten $T_A$-$q_A$ combinations at five wind speeds under either Ambient or Forced longwave conditions). The typical procedure for a given longwave forcing and air temperature-specific humidity combination in the tunnel was to begin at a wind speed of 0.5 m s$^{-1}$ (or sometimes 4 m s$^{-1}$) and then wait for the steady state

condition (typically an hour or so, see Section 4.1) before changing to the next wind speed and so on. Typically we completed the measurements for the five pre-determined wind speeds at a given temperature-specific humidity-longwave forcing combination within a single day.

To ensure reliable surface temperature measurements of the water bath using the thermal camera we avoided experiments where condensate formed on the inside of the interior film. The problem with condensate is that liquid water droplets on the film absorb most of the incoming longwave radiation (e.g., Fig. 1) but re-emit longwave radiation at the local water droplet temperature which interfered with the thermal camera measurements of the water bath. We had extensive difficulties with condensation in two evaporation sub-experiments. We were unable to complete the 35°C-25 g kg$^{-1}$ sub-experiment due to condensation repeatedly forming on the interior film at the highest wind speed. Instead we completed that sub-experiment at 35°C-20 g kg$^{-1}$ (Fig. 3). The same situation also occurred for the 45°C-35 g kg$^{-1}$ sub-experiment and we completed that sub-experiment at 45°C-30 g kg$^{-1}$ (Fig. 3). Upon completion of the measurement program, we found the most extreme evaporation sub-experiments (45°C-7 g kg$^{-1}$; 45°C-15 g kg$^{-1}$; 45°C-25 g kg$^{-1}$) failed routine quality control checks and were discarded. The final evaporation database included seven $T_A$-$q_A$ combinations (Fig. 3) at five different windspeeds (0.5, 1, 2, 3, 4 m s$^{-1}$) under two different longwave radiation forcing conditions (Ambient/Forced) giving a total of 70 individual evaporation measurements. Experiments are named using the nomenclature *Forcing-T-q-U*. For example, Ambient-T15-q7-U2 is an experiment done using the Ambient forcing (i.e., laboratory air temperature ~ 19°C) with target tunnel conditions at 15°C and 7 g kg$^{-1}$ and wind speed of 2 m s$^{-1}$. The nomenclature Forced-T15-q7-U2 refers to the same conditions but with laboratory air temperature set to 31°C.

**2.5 Summary**

In summary, the radiation calibration experiments quantified the amount of longwave radiation arriving at the water surface as a function of $T_A$-$q_A$ in the wind tunnel at two different longwave radiative forcings. Further, the evaporation experiments also held $T_A$-$q_A$ fixed in the wind tunnel and measured the response of the water bath surface ($T_S$) and bulk ($T_B$) temperature and the latent heat flux ($LE$) to a change in the longwave forcing at different wind speeds ($U$). By this construction our aim was to identify whether a prescribed longwave forcing would preferentially evaporate water and/or heat the water body. The minor complication was that not all the evaporation sub-experiments had an equivalent radiation calibration (Fig. 3; T35-q20, T45-q30) because of the above-noted problems with condensation encountered during the evaporation experiments. For that reason we chose to develop a simple radiative transfer model to quantify the radiative forcing and the development and verification of this model is described in the next section.

**3 Longwave Radiation at the Water Surface**

In this section we summarise the emissivity of various surfaces (section 3.1) and describe the underlying radiative transfer using a simple system based on one film that explicitly includes the effect of moist air within the tunnel (section 3.2). We then describe the optical properties of a single piece of film (section 3.3) and outline a simple theory for (longwave) radiative transfer through the two parallel films (Section 3.4) which is then modified to accommodate for the viewing geometry (section 3.5). The full theory for the incoming longwave radiation at the water surface is then tested (section 3.6) and then extended to

estimate the outgoing longwave radiation (and surface temperature) from the water surface (section 3.7). We conclude with a brief summary (section 3.8).

## 3.1 Emissivity of various surfaces

We used the radiometer (Kipp and Zonen: Model CNR1 net radiometer) to determine the (longwave) emissivity for several different surfaces. The process involved placing the radiometer as close as possible to an emitting source of known temperature $T$ and calculating the change in the measured outgoing radiative flux with respect to $\sigma T^4$ (with $\sigma$ the Stefan-Boltzmann Constant, Table 1) to yield the emissivity. We made extensive use of commercial waterproof paint to, for example, paint the inside of the wind tunnel, and to paint several other surfaces used in ancillary experiments. For the painted interior of the wind tunnel and other surfaces we found the emissivity to be 1. By this same approach we found the emissivity of the water surface ($\varepsilon_S$) to be 0.95 (within 0.005, results not shown). With that, the surface temperature of the evaporating water bath $T_S$ is related to the outgoing ($R_{o,S}$) and incoming ($R_{i,S}$) longwave radiative fluxes at the water surface (Fig. 2b) by,

$$R_{o,S} = \varepsilon_S \, \sigma \, T_S^4 + (1 - \varepsilon_S)R_{i,S} \qquad . \tag{3}$$

## 3.2 Radiative transfer through one film with moist air correction

We begin by describing the simplest case of longwave radiative transfer across one intervening film layer separating two black bodies in a *vacuum* (Fig. 4). For the theory we adopt the familiar grey body approximation (Sparrow and Cess, 1966, section 3-3, p. 86) with the bulk reflection ($\alpha$), absorption ($\beta$) and transmission ($\tau$) coefficients all assumed to be independent of temperature and constrained by,

$$\alpha + \beta + \tau = 1 \qquad . \tag{4}$$

By Kirchoff's law the emission from the film is given by $\beta\sigma T^4$ with the film assumed to be at the same temperature as the laboratory walls $T_L$ (Fig. 4). Hence in principle the outgoing longwave flux at the level of the thermal camera is given by the sum of transmitted ($\tau\sigma T_0^4$), emitted ($\beta\sigma T_L^4$) and reflected ($\alpha\sigma T_L^4$) components and is a "mixture" of both bounding black body temperatures ($T_0$, $T_L$). As shown below (section 3.3), with $\tau \rightarrow 1$ while $\alpha$ and $\beta$ both $\rightarrow 0$, it follows that the outgoing longwave radiative flux at the level of the thermal camera will be dominated by the transmitted component ($\tau\sigma T_0^4$). The same holds for the incoming longwave flux at the lowest level.

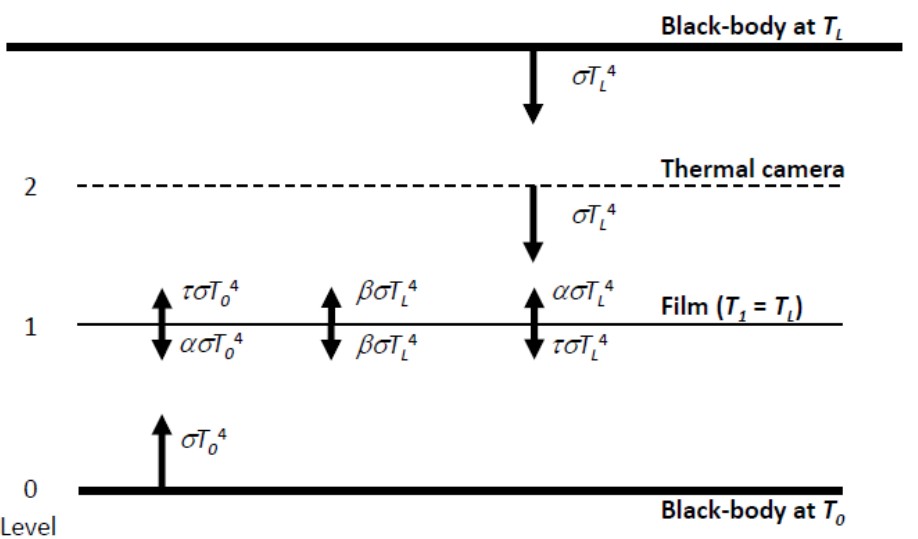

**Figure 4: Longwave radiative transfer through a single film layer between two black bodies at temperatures $T_0$ and $T_L$. The intervening space is assumed to be a vacuum with the film at the laboratory temperature $T_L$.**

In reality, the intervening space in our experiments is not a vacuum but is instead occupied by moist air. Recall that the tunnel has a 300 mm (square) cross section and over this distance we anticipate that the moist air only has a minor impact on the radiative fluxes. While minor, we found that the impact could not be ignored because offline calculations using a radiative transfer scheme (Shakespeare and Roderick, 2021) showed that the flux could vary by up to 16 W m$^{-2}$ (against a typical background of order 500 W m$^{-2}$) due to the water vapour under the most extreme situations sampled in this study. A scheme to account for the presence of moist air is outlined in Fig. 5. With reference to that figure, the black body longwave flux emitted upwards from the base is transmitted through a slab of moist air of thickness $z$ (m) having an (effective) absorptivity $\beta_A$ (m$^{-1}$). The balance not transmitted is absorbed by the radiatively active gases (i.e., the greenhouse gases) and then reemitted at the local temperature. With reference to Fig. 5, the difference $dR$ between the longwave radiation arriving at the upper level and that leaving the lower level is,

$$dR(T_0, T_L, q_L, z) = \left[\sigma T_O^4 \, e^{-\beta_A z} + \sigma T_L^4 \left(1 - e^{-\beta_A z}\right)\right] - \left[\sigma T_O^4\right] = (\sigma T_L^4 - \sigma T_O^4)(1 - e^{-\beta_A z}) \qquad . \qquad (5)$$

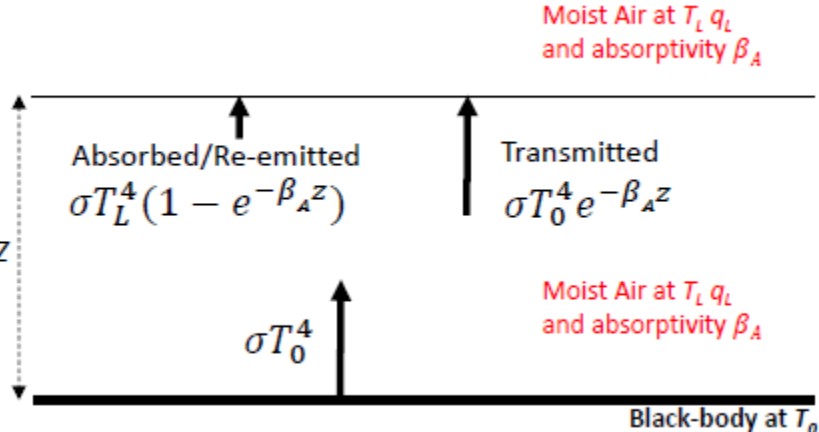

**Figure 5: Underlying principle of the moist air correction ($dR$). The fate of the emitted black body flux ($\sigma T_0^4$) passing through a slab (thickness $z$) of moist air (at $T_L$, $q_L$) having longwave absorptivity $\beta_A$.**

We used a line-by-line radiative code (Schreier et al., 2019) for the atmosphere to parameterise the longwave absorptivity for a slab of moist air at a total pressure of 1 bar (Shakespeare and Roderick, 2021) for different slab thicknesses (0.01, 0.1, 0.3 0.5 m). We found that over the thickness range considered here (0.01-0.5 m) that the absorptivity depended primarily on the specific humidity and thickness of the moist air slab according to (Appendix A),

$$\beta_A = 0.90 \, z^{-0.68} \, q^{(0.44 \, z^{-0.12})} \qquad\qquad , \qquad\qquad (6)$$

with $q$ the specific humidity (kg kg$^{-1}$) and $z$ the thickness (m) of the moist air slab. To give a numerical example, for a 0.3 m thick slab with $T_0 = 19°C$, $T_L = 45°C$ and $q_L = 0.030$ kg kg$^{-1}$, the moist air absorptivity $\beta_A$ is 0.343 m$^{-1}$ and the dimensionless optical thickness (= $\beta_A z$) is 0.102 with the final calculated $dR$ correction (per Eqn 5) equal to +16.4 W m$^{-2}$. In this numerical example, some 90% (i.e., $e^{-0.102}$) of the original black body emission (at $T_0$) is transmitted through the moist air with the remaining 10% absorbed and then re-emitted at the warmer temperature (at $T_L$) which is the origin of the positive $dR$ correction in this example. This represents the most extreme conditions encountered in this study (Fig. 3). If the moist air was instead cooler than the adjacent black body then the correction would be negative. Alternatively, if the moist air and adjacent black body were at the same temperature there is no correction irrespective of the prevailing humidity. In essence this is how the greenhouse effect operates. By comparison, if we had used the lowest moist air specific humidity used in the evaporation experiments (0.007 kg kg$^{-1}$, see Fig. 3) with the above-noted temperatures, the moist air absorptivity would be 0.164 m$^{-1}$ with the optical thickness equal to 0.049 implying that slightly more than 95% (i.e., $e^{-0.049}$) of the longwave radiation would be transmitted through a 0.3 m thick slab of moist air. These limiting cases bracket the range of values considered in this study.

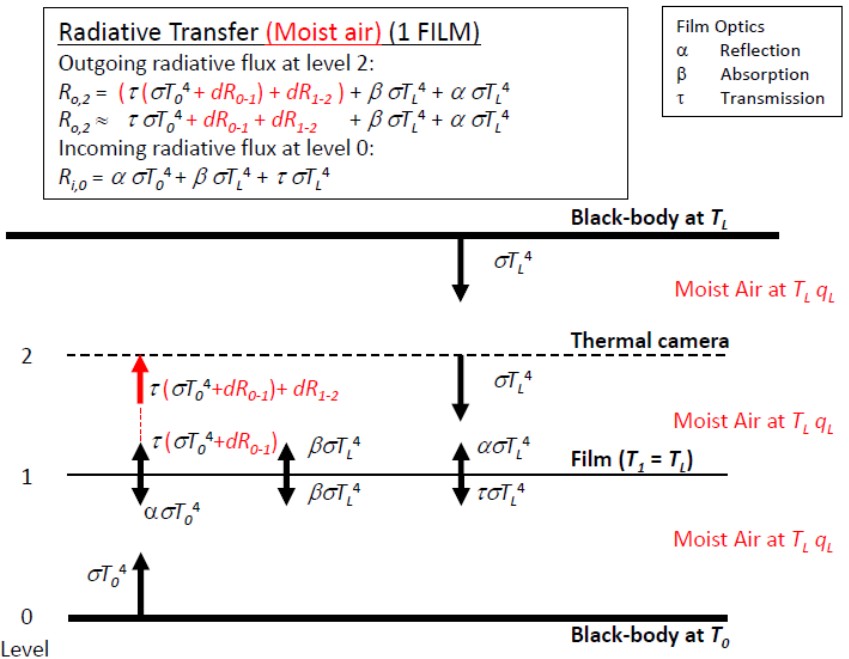

**Figure 6: Longwave radiative transfer through one film between two black bodies at temperatures $T_0$ and $T_L$ modified to account for moist air. The moist air is at the laboratory temperature $T_L$, with specific humidity $q_L$.**

We combine the moist air correction (Fig. 5) with the original transfer scheme (Fig. 4) to construct a realistic model for a single layer of film (Fig. 6). With reference to Fig. 6, the outgoing longwave flux emitted at the base ($\sigma T_0^4$) that arrives at the film is now $\sigma T_0^4 + dR_{0-1}$ with $dR_{0-1}$ denoting the change due to travelling from level 0 to level 1 because of interactions with the moist air. Some of the incident flux at level 1 is then transmitted through the film ($= \tau(\sigma T_0^4 + dR_{0-1})$) and some of that

modified flux will be absorbed and/or reflected. Again we note that with $\tau \to 1$ (hence $\alpha$ and $\beta$ both $\to 0$) (see later in Table 2) we only need consider modifications to the transmitted flux in this study. The transmitted flux is further modified when travelling through the moist air from level 1 to level 2. With $\tau \to 1$, we separate the corrections from the transmission coefficient and the outgoing longwave flux arriving at the level of the camera (= level 2 in Fig. 6, $R_{o,2}$) can be usefully approximated by,

$$R_{O,2} \approx \tau\sigma T_0^4 + dR_{0-1} + dR_{1-2} + \beta\sigma T_L^4 + \alpha\sigma T_L^4 \qquad . \qquad (7)$$

We further note that in Fig. 6, a moist air correction is not required for the incoming flux at the base ($= \alpha\sigma T_0^4 + \beta\sigma T_L^4 + \tau\sigma T_L^4$) because the temperature is uniform ($T_L$) in that direction.

### 3.3 Optical properties of the film

To estimate the bulk transmission through the film we conducted an experiment using the single film theory outlined in Fig. 6. The experiment is fully described in Appendix B. In brief, we measured the outgoing longwave radiation arriving at the thermal camera through one film layer from a known black body source whose temperature was varied over five discrete steps (10, 20, 30, 40, 50 °C) and at two different laboratory temperatures ($T_L$; 19, 31 °C) giving a total of 10 observations. By this experimental arrangement we were unable to distinguish α from β and we could only independently determine their sum. With that, the least squares results were $\tau = 0.908 \mp 0.029\ (\mp 1sd)$ and $(\alpha + \beta) = 0.092 \mp 0.032\ (\mp 1sd)$. The experimental results were in close accord with theoretical expectations (Eqn 4) with the sum of the transmission and the reflection plus absorption equal to 1 within experimental uncertainty. The results show the plastic film was highly transmissive with some 90.8% of the incident longwave radiation transmitted. Previous research has found standard polyethylene (i.e., plastic) film to be highly transmissive of longwave radiation with a bulk transmissivity of 0.75 (Koizuka and Miyamoto, 2005) to 0.76 (Horiguchi et al., 1982) reported for a film thickness of 0.1 mm. Our film was substantially thinner (0.022 mm) which would account for the higher bulk transmissivity (= 0.908) that we found experimentally. Using the experimental values for the bulk optical properties we were able to estimate the transfer of longwave radiation through a single piece of film with a typical error of 2.0 W m$^{-2}$ (Fig. B2).

To separate the reflection from the absorption we conducted an additional experiment using two plastic films (with 10 mm air gap) and altered the temperature of one film (thereby changing the emitted longwave component from that film layer) independently of the other film. The experiment is fully described in Appendix B. By again using a least squares fit we found the reflection coefficient $\alpha = 0.047$ with an overall RMSE of 3.4 W m$^{-2}$ (Fig. B3). Using Eqn 4, the implied absorption coefficient was $\beta = 0.045$. The results are summarised in Table 2.

| Variable | Value | Comment |
|---|---|---|
| $\alpha$ | 0.05 ± 0.03 (±1sd) | Bulk reflection coefficient |
| $\beta$ | 0.04 ± 0.03 (±1sd) | Bulk absorption coefficient |
| $\tau$ | 0.91 ± 0.03 (±1sd) | Bulk transmission coefficient |

Table 2   Values for bulk reflection (α), absorption (β) and transmission (τ) coefficients of a single layer of plastic film.

### 3.4 Theory for radiative transfer through two parallel films

The more general case for radiative transfer in the operational wind tunnel (Fig. 2a) with two plastic films is shown in Fig. 7. In developing this scheme we ignored any individual radiative flux with more than one reflection and/or absorption coefficient

and again we only account for moist air corrections on transmitted components. With that we note that the incoming radiative

flux at level 0 and the outgoing flux at level 3 both have five distinct terms plus the relevant moist air corrections.

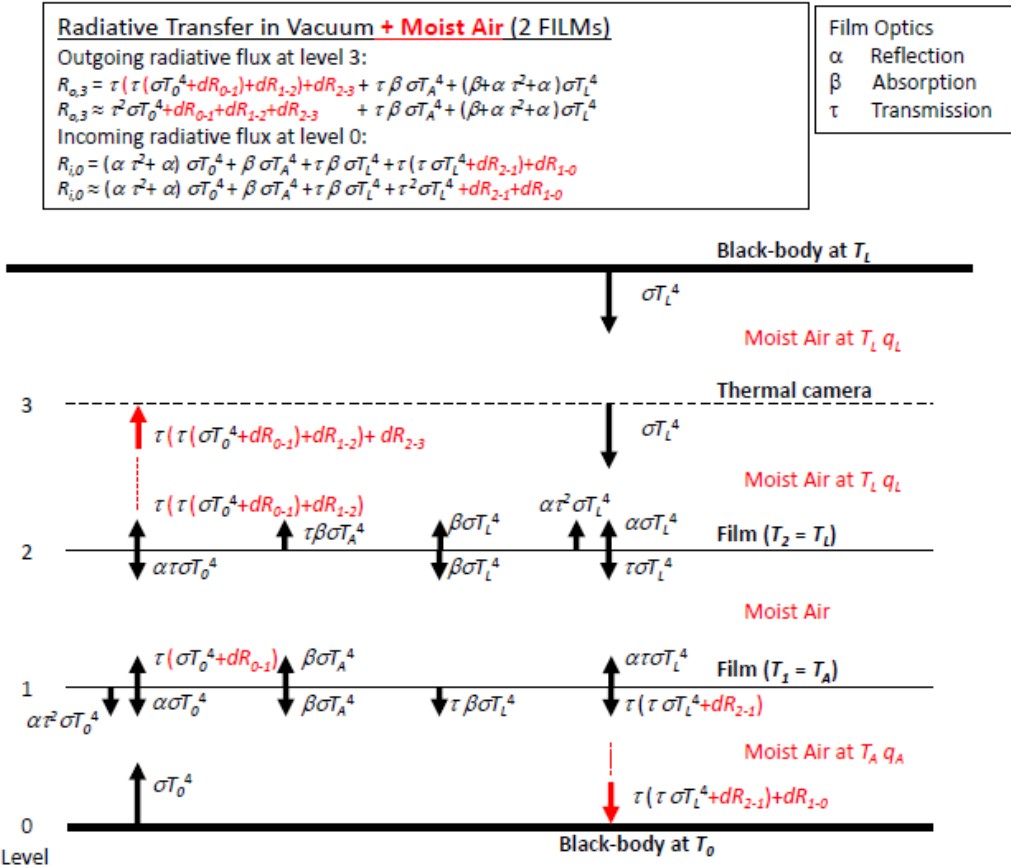

**Figure 7: Longwave radiative transfer through two films between two black bodies at temperatures $T_0$ and $T_L$. The intervening space is occupied by moist air in the tunnel ($T_A$, $q_A$) or the laboratory ($T_L$, $q_L$) with the assumed temperature of each film as noted.**

## 3.5 Modified theory to account for the viewing geometry

The previous theory to describe radiative transfer through the tunnel implicitly assumed an infinite horizontal extent (Fig. 7).

That was suitable for the experiments used to determine the bulk optical properties of the film (see Appendix B) but the

geometry of the operational tunnel configuration is more complex (Fig. 2). In the tunnel the longwave radiation arrives at the

water surface from both the film and the tunnel (Fig. 8). A further complication is that a small component of the incoming

longwave radiation is emitted from the PVC frame (assumed emissivity = 1) that holds the plastic film in place, with the PVC frame having the same temperature as the (air in the) tunnel.

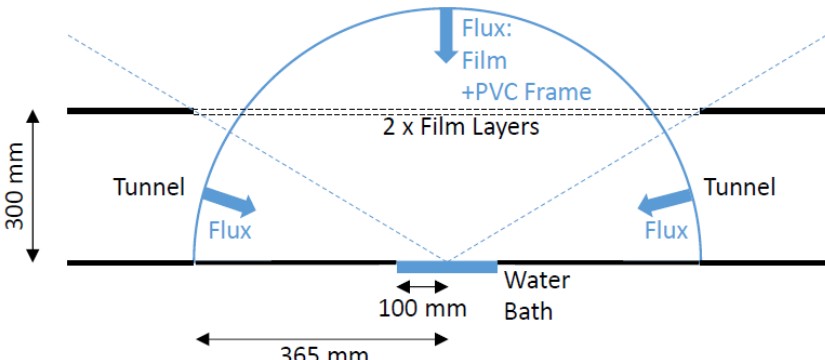


**Figure 8: Schematic drawing showing separate contributions to the incoming longwave radiation at the water surface. The diagram is a cross section along the centreline of tunnel showing the hemispherical geometry used to estimate the incoming longwave radiation at the water surface arriving from the tunnel, film and PVC frame.**

To quantify the three separate contributions to the incoming longwave radiation at the water surface, we first used three-dimensional geometry to calculate the fraction of the hemisphere occupied by the three radiation sources (tunnel, film, PVC frame). The surface area of a hemisphere with radius 0.365 m is 0.8371 m². When each separate component is projected onto that hemisphere, the surface area occupied by the film is 0.6676 m², while it was 0.1278 m² for the tunnel and 0.0417 m² for the PVC frame. Some of the radiation arrives from an acute angle and each component requires a cosine correction to calculate

the contribution to the total (i.e., when integrated over the hemisphere). This adjustment can be readily calculated for each of the three separate contributions by projecting each of the three hemispheric segments onto a circle in the horizontal plane having the same radius (Monteith and Unsworth, 2008, Fig. 4.4, p. 48). The total projected area of the hemisphere (radius 0.365 m) is 0.4185 m² with the film occupying 0.3531 m² (84.4%), the PVC frame occupying 0.0236 m² (5.6%) and the tunnel occupying 0.0418 m² (10.0%). Noting that the tunnel and PVC frame are at the temperature of the air in the tunnel ($T_A$) we can

combine those into a single term that occupies 15.6% of the projected area with the remainder (84.4%) occupied by the film.

We are now in a position to define the incoming longwave radiation at the water surface using the theory. Using $g_0$ to denote the (projected) area fraction of the tunnel plus PVC frame with both at temperature $T_A$, and taking the results from Fig. 6, we calculate the incoming radiation at the water surface ($R_{i,S}$) as,

$$R_{i,S} = g_0 \left(\sigma T_A^4\right) + (1 - g_0)\left((\alpha\tau^2 + \alpha + \beta)\sigma T_A^4 + (\tau\beta + \tau^2)\sigma T_L^4 + dR_{2-1} + dR_{1-0}\right) \qquad , \qquad (8a)$$

with the moist air corrections calculated using $dR_{2-1}(T_L, T_A, q_A, 0.01)$ and $dR_{1-0}(T_L, T_A, q_A, 0.30)$. With $g_0$ set to the theoretically calculated value (= 0.156), and using the experimental values for the bulk optical properties (Table 2), we derive the following theory-based equation,

$$R_{i,S} = 0.156\,(\sigma T_A^4) + 0.844\,(0.1314\,\sigma T_A^4 + 0.8645\,\sigma T_L^4 + dR_{2-1} + dR_{1-0}) \qquad\qquad , \qquad\qquad (8b)$$

to predict the incoming longwave radiation at the water surface. From this equation we see that $R_{i,S}$ is a "mixture" mostly determined by $T_L$ with a smaller contribution from $T_A$ and minor contributions from two moist air adjustments. This theory is tested by experiment in the following section.

## 3.6 Incoming longwave radiation at the water surface

We evaluate the theory (Eqn 8a) using measurements made in the previously described radiation calibration experiments (n =
20, i.e., ten $T_A$-$q_A$ combinations under both the Ambient and Forced longwave conditions, see Fig. 3) in Fig. 9. The results using theory plus the experimentally determined bulk optical properties ($\alpha$, $\beta$, $\tau$) are excellent with an overall RMSE of 3.1 W m$^{-2}$ (Fig. 9a). This RMSE was slightly greater than the original RMSE (2.0 W m$^{-2}$) reported when estimating the transmission through the film (Fig. B2). Close visual inspection of Fig. 9a reveals that the slopes for the ambient ($T_L$ = 19°C) and forced ($T_L$ = 31°C) data are both slightly greater than 1 implying a slight but consistent bias in the results. That is not surprising. For
example, both the radiative transfer and the geometric derivation of the projected area fraction parameter $g_0$ (= 0.156) implicitly assumed isotropic radiation at every step of the derivation but we expect slight errors in that assumption. Hence, we also calculated the numerical value of the geometric parameter $g_0$ that had the minimum RMSE (= 2.2 W m$^{-2}$) which also removed the above-noted bias (Fig. 9b). We subsequently used the tuned value ($g_0$ = 0.128) in Eqn 8a to calculate the incoming longwave radiation at the water surface for each of the evaporation experiments (n = 70).


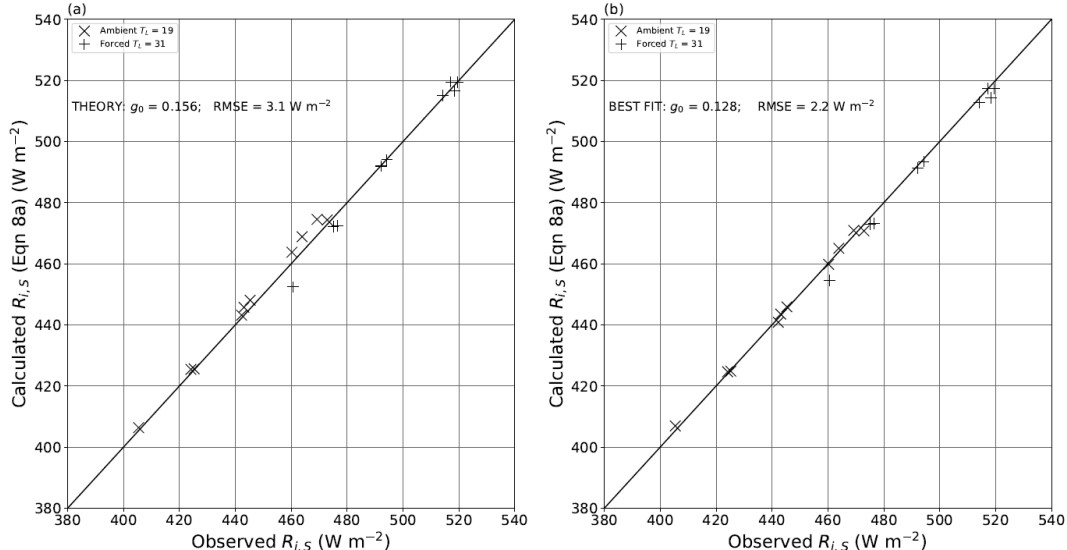

**Figure 9: Comparison of theoretical (Eqn 8a) and observed incoming longwave radiation at the water surface. (a) Uses $g_0$ = 0.156 as per theory (Linear regression: y = 0.9855 x + 7.3, $R^2$ = 0.991, RMSE = 3.1 W m$^{-2}$, n=20). (b) Tuned to locate the value of $g_0$ (=0.128) with the lowest RMSE (Linear regression: y = 0.9772 x + 9.7, $R^2$ = 0.997, RMSE = 2.2 W m$^{-2}$, n=20). Full lines are 1:1.**

### 3.7 Outgoing longwave radiation from the water surface

The transfer of outgoing longwave radiation from the water surface through the moist air and film layers before arrival at the thermal camera follows the same basic theory (Fig. 7, Fig. 8) and is a function of the prevailing temperatures ($T_S$, $T_A$, $T_L$), bulk optical properties ($\alpha$, $\beta$, $\tau$) and the overall geometry of the camera-tunnel system. By inspection of Fig. 7 and Fig. 8, we used a new (but analogous) geometric parameter, $g_1$, to calculate the outgoing longwave radiation arriving at the thermal camera from the water surface ($R_{o,C,S}$),

$$R_{o,C,S} = g_1 \left(\sigma T_L^4\right) + (1 - g_1) \left(\tau^2 R_{o,S} + \tau\beta\sigma T_A^4 + (\beta + \alpha\tau^2 + \alpha)\sigma T_L^4 + dR_{0-1} + dR_{1-2} + dR_{2-3}\right) \qquad , \qquad (9)$$

with $R_{o,S}$ (Fig. 2b) the outgoing longwave radiation from the water surface. Eqn 9 can be re-arranged to derive the required expression for $R_{o,S}$,

$$R_{o,S} = \frac{1}{\tau^2}\left[\left(\frac{R_{o,C,S} - g_1\sigma T_L^4}{1 - g_1}\right) - \tau\beta\sigma T_A^4 - (\beta + \alpha\tau^2 + \alpha)\sigma T_L^4 - dR_{0-1} - dR_{1-2} - dR_{2-3}\right] \qquad . \qquad (10a)$$

All quantities on the right hand side of Eqn 10a are measured/known with the exception of the geometric parameter $g_1$. In the evaporation experiments, the thermal camera used to infer $T_S$ (temperature of the evaporating surface) was mounted in an off-vertical position (Fig. 2a) and we were unable to use simple theory to calculate the geometric factor ($g_1$). Instead we used a semi-empirical approach to quantify the geometric parameter. During the evaporation experiments we simultaneously recorded the longwave radiation arriving at the thermal camera from the water surface and from the camera calibration spot whose

temperature had also been measured independently via a thermocouple ($T_T$, Fig. 2b). We used those two camera calibration spot measurements embedded within the evaporation experiments (n = 70) to derive an empirical value for $g_l$ (= 0.160). The approach is fully described elsewhere with an estimated error in the outgoing longwave radiative flux from the water surface of 2.9 W m$^{-2}$ (Appendix C).

With the relevant numerical values ($g_l$ = 0.160, bulk optical properties from Table 2), we have,

$$R_{o,S} = 1.2076 \left[ \left( \frac{R_{o,C,S} - 0.160\, \sigma T_L^4}{0.840} \right) - 0.0364\, \sigma T_A^4 - 0.1314\, \sigma T_L^4 - dR_{0-1} - dR_{1-2} - dR_{2-3} \right] \qquad . \qquad (10b)$$

With $R_{o,S}$ calculated we rearrange Eqn 3 to calculate the surface temperature,

$$T_S = \left( \frac{R_{o,S} - (1 - \varepsilon_S) R_{i,S}}{\sigma\, \varepsilon_S} \right)^{1/4} \qquad , \qquad (11)$$

using the experimentally measured emissivity for the water surface ($\varepsilon_S$ = 0.95). The relevant moist air corrections (in Eqn 10ab) are given by $dR_{0-1}(T_S, T_A, q_A, 0.44)$, $dR_{1-2}(T_S, T_A, q_A, 0.015)$ and $dR_{2-3}(T_S, T_L, q_L, 0.125)$ which presents two complications. The first is that the moist air corrections were derived assuming a black body but the water surface is not a black body. However it is sufficiently close ($\varepsilon_S$ = 0.95) for that complication to be safely ignored (results verified but not shown). The second complication is that when $R_{o,S}$ is first calculated, the surface temperature of the water surface $T_S$ is unknown but it is needed to calculate the moist air corrections. We used an iterative approach with the first iteration using the measured bulk water temperature $T_B$ (Fig. 2b) as an initial estimate for $T_S$ in each of the moist air corrections. After the first iteration we used the now updated value of $T_S$ to re-calculate the moist air corrections and hence update the final solution for $R_{o,S}$ and $T_S$. One iteration was sufficient for convergence of the calculation under all conditions. The surface temperature estimates are compared with the directly measured bulk water temperatures in a subsequent section (section 4.3).

## 3.8 Summary

We have developed theory (Fig. 7, Fig. 8) that predicts the measured incoming longwave radiation at the water surface (Eqn 8a) with an error of around 3.1 W m$^{-2}$ (Fig. 9a). With a very small empirical modification to the theory that error was reduced to 2.2 W m$^{-2}$ (Fig. 9b). We used the same theory supplemented with one empirically determined geometric parameter to predict the outgoing longwave radiation from the water surface using thermal camera measurements with an estimated error of 2.9 W m$^{-2}$ (Fig. C1b). Under the prevailing conditions that is equivalent to an error in the surface temperature of ~ 0.5°C. We use these error estimates (±1sd) in subsequent sections to evaluate the suitability of the experiments to achieve the aims of the project.

**4 Evaporation from the Water Surface**

In this section we first describe the approach to steady state evaporation (section 4.1) and characterise the variability in the key experimentally controlled variables once at steady state (section 4.2). We then compare the direct measurements of the bulk water temperature with the surface temperature measurements made using the thermal camera (section 4.3), briefly examine how the evaporation and water temperature respond to windspeed (section 4.4) and compare the water bath temperatures (surface and bulk) with the theoretical wet bulb temperature (section 4.5). We conclude with a brief summary (section 4.6).

**4.1 Approach to steady state evaporation**

During the experiments we found that the initial evaporation would vary depending on the initial temperature of water placed in the bath before finally coming to a stable steady state when the water bath temperature also stabilised. In all evaporation experiments (n = 70) we waited sufficient time for the steady state to occur and measured the variables by taking their average during the steady state period.

To demonstrate the underlying principle we conducted two experiments to demonstrate the approach to steady state under the same externally imposed conditions (Ambient-T35-q25-U2; tunnel air temperature of 35°C, specific humidity of 25 g kg$^{-1}$ and wind speed of 2 m s$^{-1}$). Figure 10 depicts the first experiment which was begun by placing water at 15°C in the water bath. The mean air temperature in the laboratory was ~ 19°C (i.e., the 'Ambient' condition) and varied with an amplitude of ~ 1°C over a period that was ~ 900 s (i.e., 15 mins) in this example (Fig. 10a). This periodic variation was a consequence of the cooling control system deployed in the temperature controlled laboratory whose settings could not be altered. This laboratory period was not fixed since it varied with the external weather conditions. Despite that laboratory periodicity, the air temperature within the tunnel was controlled within a much tighter range and was held close to the target temperature of 35°C over the entire time period ($T_{AU}$, $T_{AD}$ in Fig. 10a) as was the wind speed (Fig. 10d). Similarly, the specific humidity of air in the laboratory also showed the same periodic behaviour (period ~ 900 s, see $q_L$ in Fig. 10c), but again, the specific humidity of air in the tunnel was controlled within a much tighter range ($q_A$, Fig. 10c). The incoming longwave radiation at the top of the tunnel was measured directly using the radiometer ($R_{i,F2}$, Fig. 10e) and also varied over the same 900 s period. The direct measurement of $R_{i,F2}$ was very close to the theoretical black body radiation at the temperature of the laboratory air as expected (see blue line in Fig. 10e). To account for the laboratory periodicity we (i) always selected the steady state time extent to be (substantially) longer than the 900 s period and we (ii) tried to define wherever possible the steady state period to be an (approximate) integer multiple of the period which largely removed/minimised the effect of laboratory periodicity.

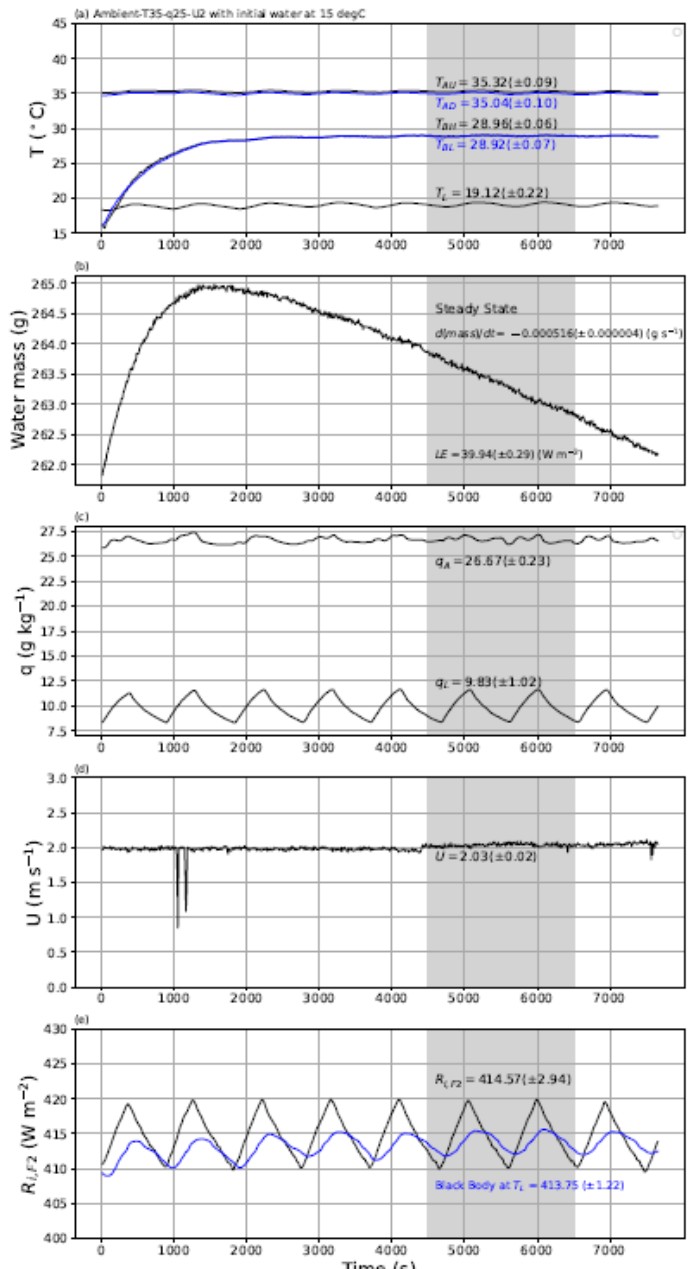

**Figure 10: An experimental demonstration of the approach to steady state. The experiment began with water at 15°C in the shallow water bath ($T_{BH}$, $T_{BL}$ in (a)) with target conditions for air in the tunnel set to Ambient-T35-q25-U2. The plots document the approach to steady state (4500-6500 s) for the evolution of (a) temperature of air in the tunnel ($T_{AU}$ (black), $T_{AD}$ (blue)), temperature of water in the shallow water bath ($T_{BH}$ (black), $T_{BL}$ (blue)) and temperature of air in the laboratory ($T_L$), (b) mass of water in shallow water bath with calculated rate of change (via linear regression) and the associated latent heat flux ($LE$), (c) specific humidity of air in the tunnel ($q_A$) and in the laboratory ($q_L$), (d) wind speed ($U$) and (e) the measured incoming longwave radiation at the top of tunnel ($R_{i,F2}$) compared with theoretical black body radiation at laboratory air temperature ($T_L$, blue). The numbers on each panel indicate the steady state averages ($\pm$ 1sd).**

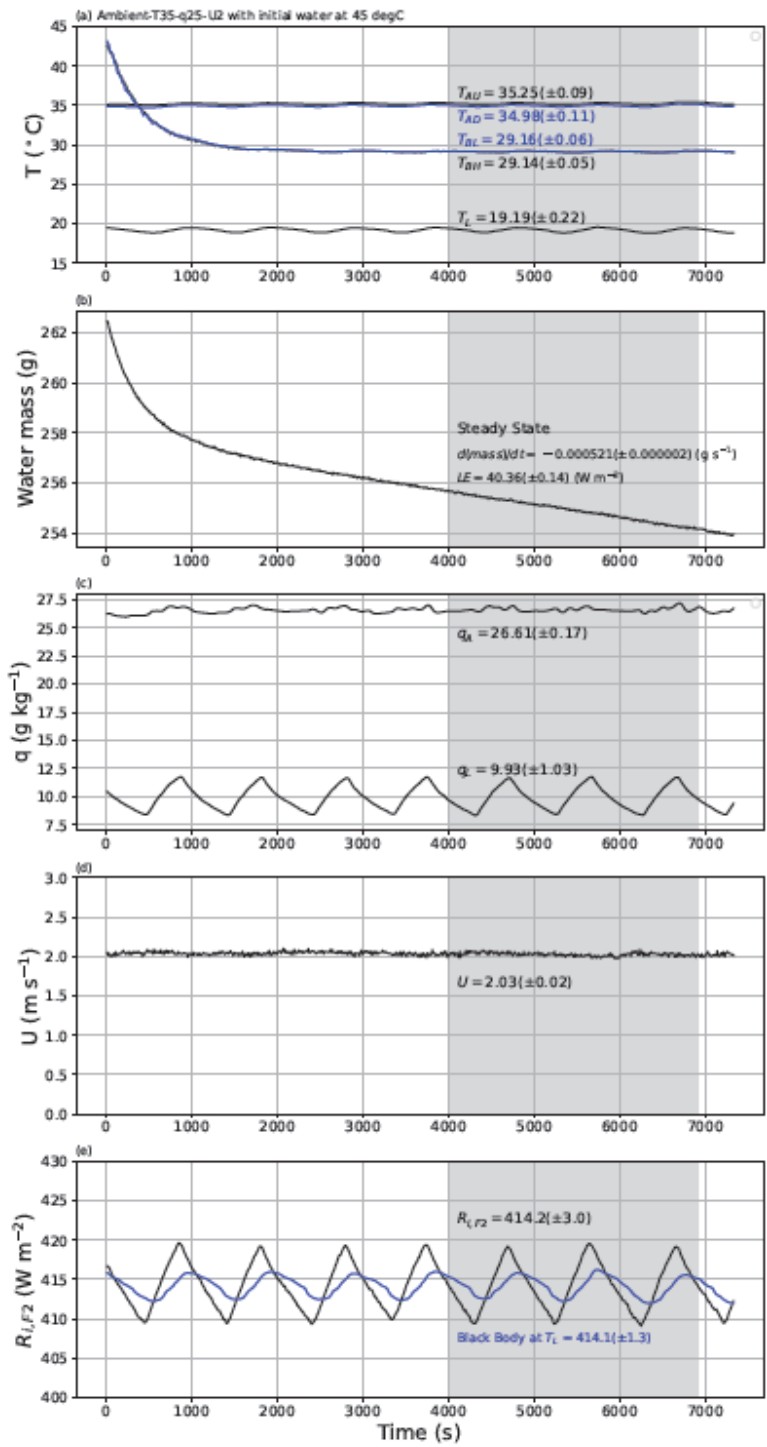

**Figure 11: Same as Figure 10 but starting with water at 45°C in the shallow water bath.**

Of most interest here is the approach to steady state in terms of the evaporation (Fig. 10b) and the water temperature in the shallow water bath (Fig. 10a). Note that this first experiment was initialised with ~ 15°C water in the shallow water bath (Fig. 10a, $T_{BH}$, $T_{BL}$). Inspection of Fig. 10a shows that the temperature of water in the shallow water bath increased at an exponentially decreasing rate towards a steady state some 4500 s from the beginning. With the initial conditions having colder water in the shallow water bath (15°C) than in the tunnel air (35°C), the initial evaporation rate was negative (i.e., condensation occurred) for the first 1500 s with a steady state evaporation rate being reached around 3000 s after the beginning of the experiment. We repeatedly observed that the time taken to reach a steady state for evaporation was slightly shorter than the time taken for the temperature of bulk water in the shallow water bath to reach steady state. Once at steady state, we calculated averages for all variables using the same user-specified time interval. Recall that the instruments were all sampled at 30 Hz and then averaged to successive 10 s periods. Hence for this example experiment, the steady state average was calculated using 201 samples (i.e., (6500-4500)/10 + 1) and the standard deviation of each measurement was also calculated using those same 201 samples.

At steady state, the bulk water in the shallow water bath had a near-uniform temperature as anticipated ($T_{BH}$ and $T_{BL}$ in Fig. 10a). Accordingly, we characterised the steady state water bath temperature ($T_B$ in Fig. 2b) as the average over the two depths. In this particular experiment we note that the steady state air temperature in the tunnel was slightly warmer in the upstream location ($T_{AU}$) relative to the downstream location ($T_{AD}$) by ~ 0.3°C (Fig. 10a). This was expected since the upstream air was closer to the radiator with the air then passing through the non-insulated part of the tunnel (i.e., the part covered with plastic film above the shallow water bath, see Fig. 2a) before entering the insulated tunnel again where the downstream air temperature was measured ($T_{AD}$ in Fig. 2b). We noted that the upstream tunnel air ($T_{AU}$) was very slightly warmer (colder) than the downstream tunnel air ($T_{AD}$) when the air in the tunnel was warmer (colder) than air in the laboratory ($T_L$) (results not shown). In other words, the part of the wind tunnel directly below the film was not quite adiabatic because the design facilitated longwave radiative exchange between the tunnel and the surroundings. With that understanding we characterised the steady state tunnel air temperature immediately above the shallow water bath ($T_A$ in Fig. 2b) as the average of the measured upstream and downstream values.

We repeated the first experiment but this time we started with water at an initial temperature of ~ 45°C in the shallow water bath (Fig. 11). This second experiment shows that the initial evaporation rate was greater than the final steady state evaporation rate (Fig. 11b) while the water in the shallow bath progressively cooled to a final steady state temperature reached some 4000 s after the beginning of the experiment (Fig. 11a). Again, at steady state the temperature of bulk water in the shallow water bath was uniform within measurement uncertainty ($T_{BH}$ and $T_{BL}$ in Fig. 11a). Importantly, the final steady state water bath temperature was more or less the same (Fig. 11a; $T_B$ = 29.15 (±0.06) °C) as in the earlier experiment (Fig. 10a; $T_B$ = 28.94

(±0.07) °C) despite the large difference in the initial temperature of water in the shallow water bath. Similarly, the steady state latent heat flux was also the same (Fig. 11b; $LE = 40.36$ (±0.14) W m$^{-2}$) as in the earlier experiment (Fig. 10b; 39.94 (±0.29) W m$^{-2}$) within measurement uncertainty. We show later (section 4.5) that this repeatable steady state occurs because the water bath has a preferred steady state temperature that is approximately equivalent to the theoretical thermodynamic wet bulb temperature.

## 4.2 Variability during the steady state period

The precision of the measurements depends on the intrinsic characteristics of the instruments and temporal variability during the designated steady state period. Over all 70 evaporation experiments, the length of the steady state period varied from 850 to 3300 s (~ 14 to 55 minutes). As noted previously, to minimise the impact of the periodic variation in $T_L$ (Fig. 10a, 11a) we (visually) selected the start and end times of the steady state to be an integer multiple of the period wherever possible (e.g., Figs 10, 11). Overall we found temporal variability during the steady state period to be the dominant source of uncertainty in the steady state averages. To summarise that uncertainty, we show the standard deviation calculated during the steady state period for six key variables across all of the 70 evaporation experiments (Fig. 12). The larger range in standard deviation for the steady state temperature of laboratory air ($T_L$, Fig. 12a) compared to that for the tunnel air ($T_A$, Fig. 12b) and the water bath ($T_B$, Fig. 12c) is consistent with the more tightly controlled temperature conditions within the wind tunnel relative to the surrounding laboratory. At steady state the tunnel air specific humidity was tightly controlled with the standard deviation less than 0.4 g kg$^{-1}$ in 67 out of 70 evaporation experiments (Fig. 12d). The wind speed remained very tightly controlled (Fig. 12e).

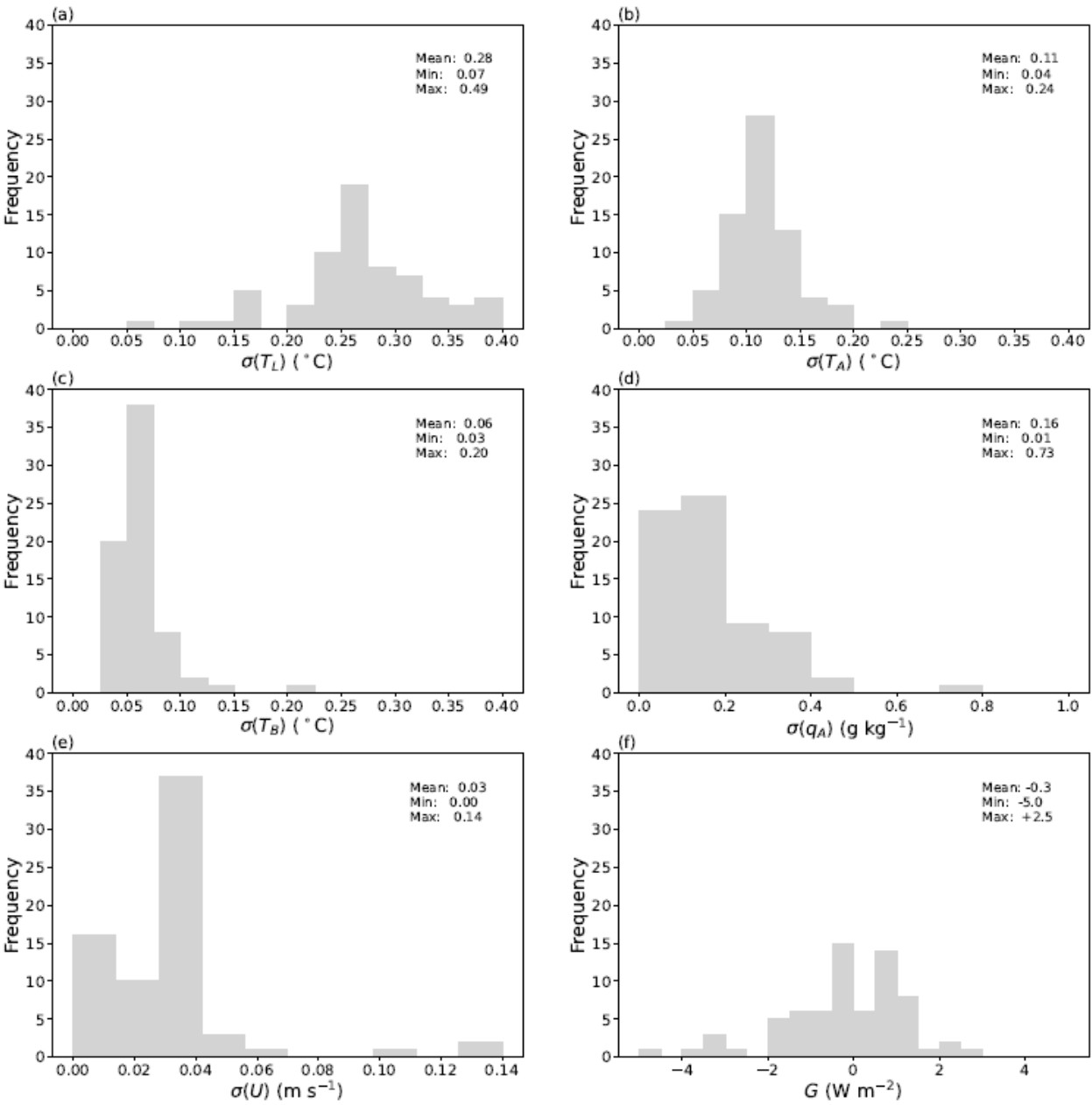

**Figure 12: Steady state variability of six key variables. Histograms show the standard deviation ($\sigma$) of measurements during the steady state period for all evaporation experiments (n = 70). Air temperature in the (a) laboratory ($T_L$) and the (b) wind tunnel ($T_A$), (c) bulk water temperature in the water bath ($T_B$) (d) specific humidity of air in the wind tunnel ($q_A$), (e) wind speed in the tunnel ($U$) and (f) the rate of change of enthalpy in the water bath ($G$).**

A very general overview of variability during the steady state period can be obtained by calculating the rate of heat storage (i.e., enthalpy flux) in the shallow water bath. We calculated the change in enthalpy of the water mass in the bath over the steady state time period using the difference between the averages of the last ten temperature measurements and the first ten measurements of the bulk water temperature ($T_B$). Dividing that enthalpy difference by the duration of the steady state time period and by the surface area of the water surface we have the equivalent rate of heat storage in the shallow water bath denoted $G$ (W m$^{-2}$). The results over all 70 evaporation experiments show that $G$ ranged from -5.0 to +2.5 W m$^{-2}$ with an overall mean very close to zero (Fig. 12f). Hence we tentatively conclude that we were able to achieve a reliable steady state in the evaporation experiments.

## 4.3 Comparing the surface and bulk water temperature

We did not have an independent measure of the surface temperature of the water bath and instead we compare it with the direct thermocouple-based measurements of the steady state bulk water temperature $T_B$ over all ($n$ = 70) evaporation experiments (Fig. 13). While the measurement approaches are completely different (thermocouple for $T_B$ and thermal camera for $T_S$), the results show a coherent relationship between the surface and bulk water temperatures under both ambient and forced conditions over the entire range of imposed conditions. Counter-intuitively, for a given $T_B$, $T_S$ is universally *colder under the forced condition* by ~ 1.2°C. Further close inspection of the ambient results reveals that $T_S > T_B$ for $T_B > 19.2$°C with 19.2°C defined empirically as that temperature where the linear regression crosses the 1:1 line (and calculated using the linear regression results in the Fig. 13 caption, i.e., 2.608/(1.136 - 1) = 19.2°C). Similarly, $T_S < T_B$ for $T_B < 19.2$°C. The same pattern holds for the forced data but with a cross-over temperature at 29.6°C. The cross-over temperatures are more or less the same as the laboratory temperature under ambient ($T_L$ ~ 19°C) and forced ($T_L$ ~ 31°C) conditions and we show later that this occurs because the wind tunnel permits longwave radiative exchange and is therefore not quite adiabatic.

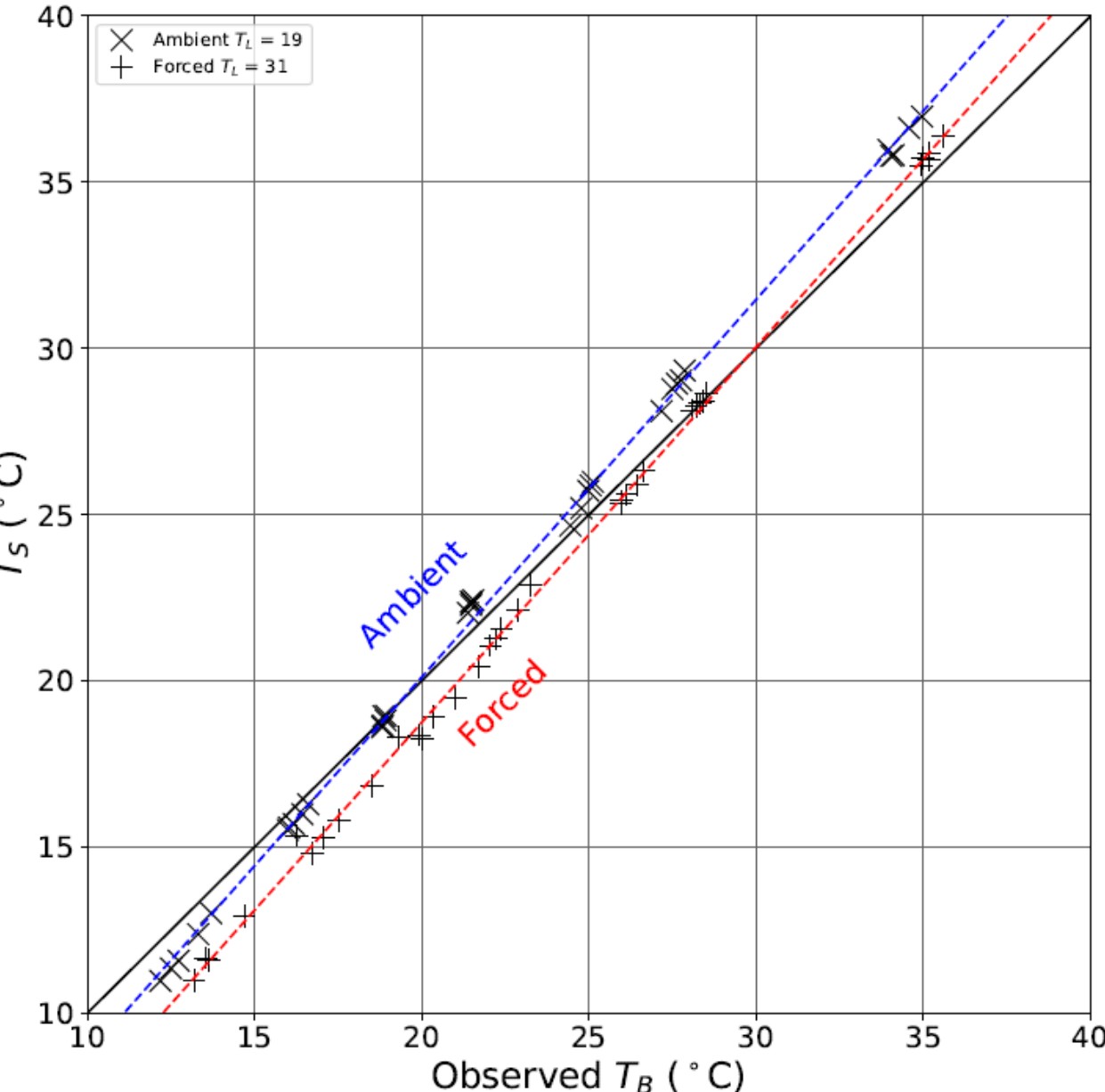

**Figure 13: Comparison of observed bulk water temperature ($T_B$) with calculated surface temperature of the evaporating water bath ($T_S$) during all evaporation experiments (n = 70). Full line is 1:1. Linear regressions for ambient (blue dashed line, y = 1.136 x − 2.608, $R^2$ = 0.999, RMSE = 1.1°C, n = 35) and forced (red dashed line, y = 1.130 x − 3.851, $R^2$ = 0.999, RMSE = 1.2°C, n = 35) conditions also shown.**


## 4.4 Typical response of evaporation and water temperature to windspeed

One key aspect of the experiment was to document how the (steady state) evaporation rate and water temperature in the shallow water bath responded to wind speed. To gain an initial overview we use data from the two most extreme laboratory experiments (Fig. 14). Briefly, the latent heat flux (and hence evaporation rate) increased (in a saturating manner) with wind speed in all experiments in a similar manner to the results depicted here (Fig. 14a). In contrast the (surface and bulk) temperature of water in the water bath increased slightly with wind speed in some experiments (e.g., Ambient-T45-q30 in Fig. 14b) but decreased

slightly in other experiments (e.g., Ambient-T15-q7 in Fig. 14b). The main point to be emphasised here is that the evaporation rate increased markedly with $U$ (as expected) in all experiments but the water temperature response was more complex with some experiments showing slight cooling while others showed slight warming with wind speed.

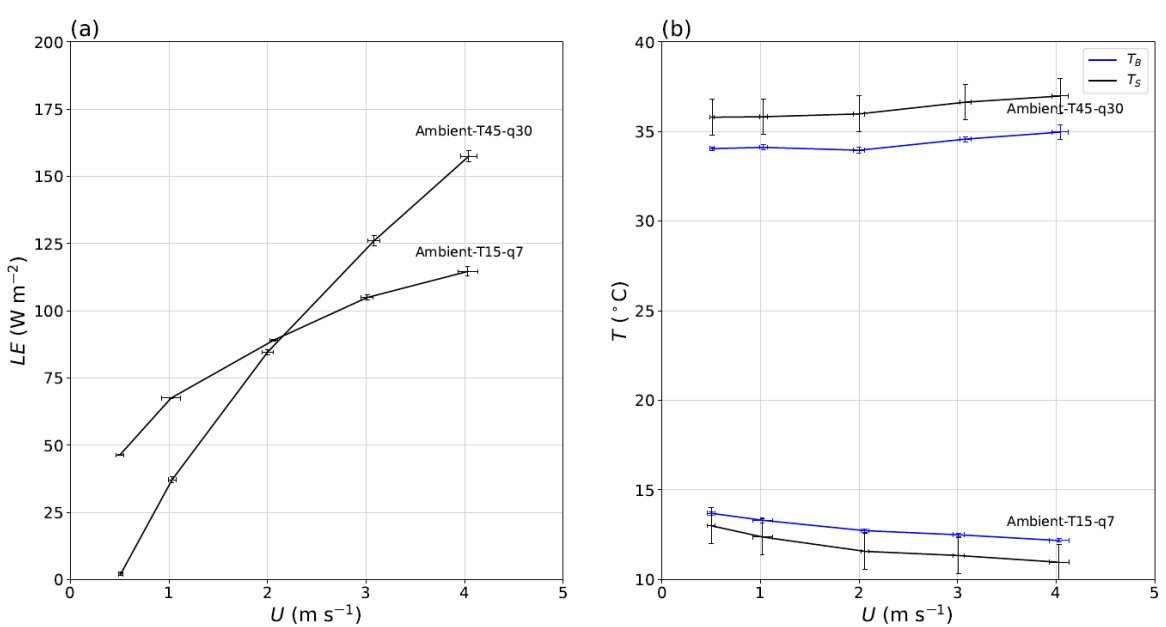

**Figure 14: Response of the steady state (a) latent heat flux ($LE$) and (b) water temperature ($T_S$, $T_B$) to wind speed ($U$) in two typical evaporation experiments. The error bars denote ±2sd (i.e., 95% confidence interval).**

## 4.5 The water bath and the theoretical wet bulb temperature

As noted previously, the final steady state evaporation and temperature of water in the water bath were independent of the

initial water temperature of water (section 4.1). In essence our shallow water bath operates as an approximate wet bulb thermometer. The concept of the wet bulb temperature assumes a closed adiabatic system containing moist air and a source of liquid water. In the adiabatic enclosure, the heat required to change the moisture content of the air (i.e., latent heat) is taken as

sensible heat from the moist air but the sum of the latent and sensible heat remains constant (Monteith and Unsworth, 2008). Hence any increase (decrease) in moisture content results in a decrease (increase) in air temperature but the overall enthalpy

remains constant. The theoretical wet bulb temperature ($T_W$) is the temperature when the moist air becomes saturated under the adiabatic constraint. In our experiment, holding $T_A$, $q_A$ constant is equivalent to holding the enthalpy constant. Given that the water bath in our experiment is 'saturated', we expect the temperature of that water bath would be approximately equal to $T_W$ after sufficient time has elapsed for a steady state to become established. Using $e$ as the symbol for vapour pressure, $T_W$ is related to $T_A$, $e_A$ by the following equation (Monteith and Unsworth, 2008),

$$e_W = e_A + \gamma(T_A - T_W)     ,\tag{12}$$

with $e_W$ (Pa) the saturation vapour pressure at $T_W$ (i.e., $e_W = e_{sat}(T_W)$) and $\gamma$ (Pa K$^{-1}$) the (so-called) psychrometer constant. Here we set $\gamma = 68$ Pa K$^{-1}$ (Appendix D) and adopt a standard saturation vapour pressure-temperature relation (Huang, 2018) to numerically solve for $T_W$ (and hence $e_W$) given $T_A$, $e_A$ and $\gamma$.

We first calculate $T_W$ for each of the seven temperature-humidity combinations using experiments conducted under ambient conditions at a wind speed of 2 m s$^{-1}$ (Fig. 15). It is immediately clear that $T_B$ is very similar to the theoretical $T_W$ in all experiments. In this example, the difference between $T_B$ and $T_W$ varies from -1.3°C to 1.3°C and is on average (= -0.3°C) very close to zero. Differences between $T_W$ and $T_B$ are expected, because as noted previously, the experimental system was not designed to be adiabatic, i.e., it has a (non-adiabatic) plastic film section that allows us to alter the incoming longwave radiation

independently of conditions inside the tunnel. Note that for experiment T35-q7, the wet bulb temperature $T_W$ is ~ 19°C which is very close to the laboratory temperature under ambient conditions ($T_L$ ~ 19°C) and we expect that this experiment should very closely approximate adiabatic conditions. Hence we also find $T_W$ ~ $T_B$ for this particular experiment. For $T_W > 19$°C we note that $T_B$ is typically less than $T_W$ while the reverse holds for $T_W < 19$°C. This is the same basic phenomenon that was noted previously (section 4.3).


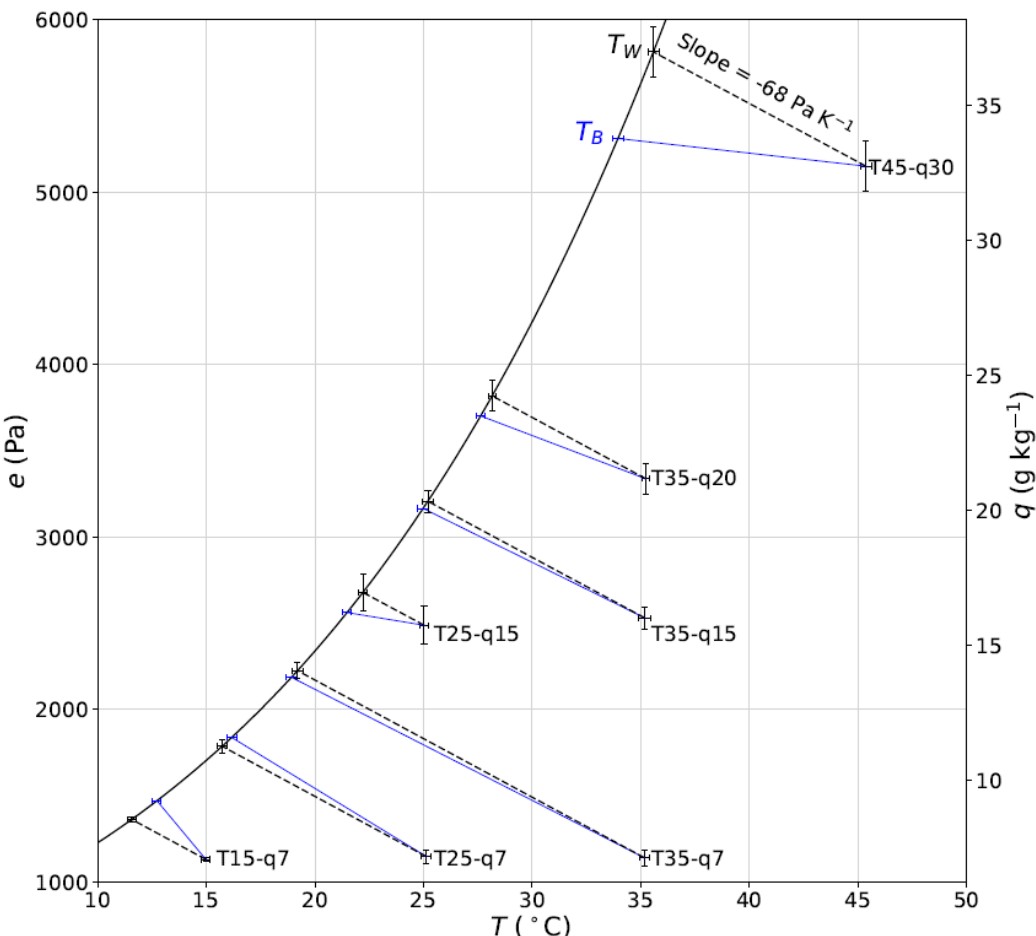

**Figure 15: Comparison of observed water bath temperature ($T_B$) with the thermodynamic wet bulb temperature ($T_W$). Plot uses all experimental data at a wind speed of 2 m s$^{-1}$ under the ambient forcing (n = 7, T15-q7, T25-q7, T35-q7, T25-q15, T35-q15, T35-q20, T45-q30). The dashed black lines join the measured air properties ($T_A$, $q_A$) to the calculated wet bulb temperature ($T_W$). The full blue lines link with the measured bulk water temperature ($T_B$). The error bars denote ±2sd (i.e., 95% confidence interval). Note that we use the same error bars for $T_W$ as for $T_A$.**

To investigate in more detail we compare $T_W$ with both $T_B$ (Fig. 16a) and $T_S$ (Fig. 16b) over all evaporation (n = 70) experiments. The same general relations found previously (Fig. 13) are also found here. For example, under the ambient condition ($T_L$ ~ 19°C), we have $T_B > T_W$ for $T_W < T_L$ and $T_B < T_W$ for $T_W > T_L$ (Fig. 16a). The same relation holds for $T_S$ (Fig. 16b) and for the forced condition ($T_L$ ~ 31°C) as well. In summary, when the wind tunnel most closely approximates an adiabatic system (i.e., $T_W$ ~ $T_L$) we find that both $T_S$ and $T_B$ closely approximate $T_W$. Interestingly, we also find that overall, $T_S$ is slightly closer to $T_W$ (Fig. 16b; RMSE ~ 0.9°C) than is $T_B$ (Fig. 16a; RMSE 1.3°C) in our experiments.

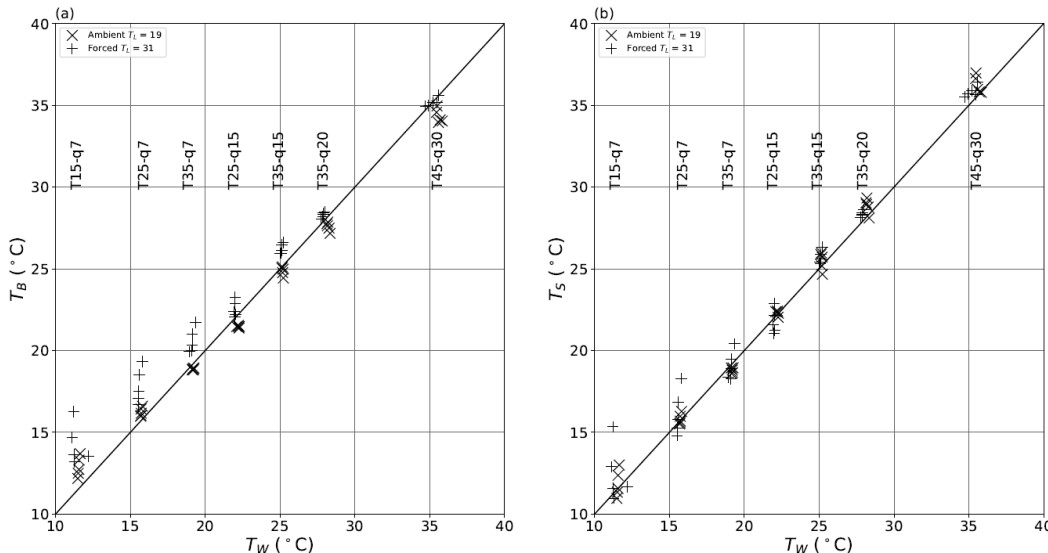

Figure 16: Comparison of theoretical wet bulb temperature ($T_W$) with the (a) bulk water ($T_B$) and (b) surface ($T_S$) temperature across all 70 evaporation experiments under ambient (×) and forced (+) conditions. The seven vertical 'clumps' of data represent the seven *T-q* combinations (as shown by vertical text labels) used in the evaporation experiments.

### 4.6 Summary

The air temperature, humidity and wind speed were successfully controlled within the experimental wind tunnel system. We found that the shallow water bath has a preferred steady state temperature that closely approximates the theoretical wet bulb temperature. That approximation is very close under adiabatic conditions when the surface temperature also very closely approximates the bulk water temperature. The preferred steady state temperature of the water bath is also associated with a repeatable steady state evaporation rate.

### 5 Magnitude of the Radiative Forcing Relative to Measurement Accuracy of *LE*

In this section we synthesise the main results from sections 3 and 4 to assess whether the experiment is sufficiently accurate to support the aims.

We begin by rewriting Eqn 2 to express the energy balance for each experiment as,

$$R_{i,S} = G + R_{o,S} + LE + H \qquad . \qquad (13)$$

For experiments at a given T-q-U combination we take the difference between the forced and ambient conditions (n=35) as follows,

$$\Delta R_{i,S} = \Delta G + \Delta R_{o,S} + \Delta(LE) + \Delta H \qquad , \tag{14}$$

with $\Delta R_{i,S}$ (Forced – Ambient) the experimentally imposed longwave radiative forcing at the water surface, $\Delta G$ the difference in the rate of enthalpy storage, $\Delta R_{o,S}$ the difference in outgoing longwave radiation from the water surface, $\Delta(LE)$ the difference in latent heat flux and $\Delta H$ the (unmeasured) difference in sensible heat flux.

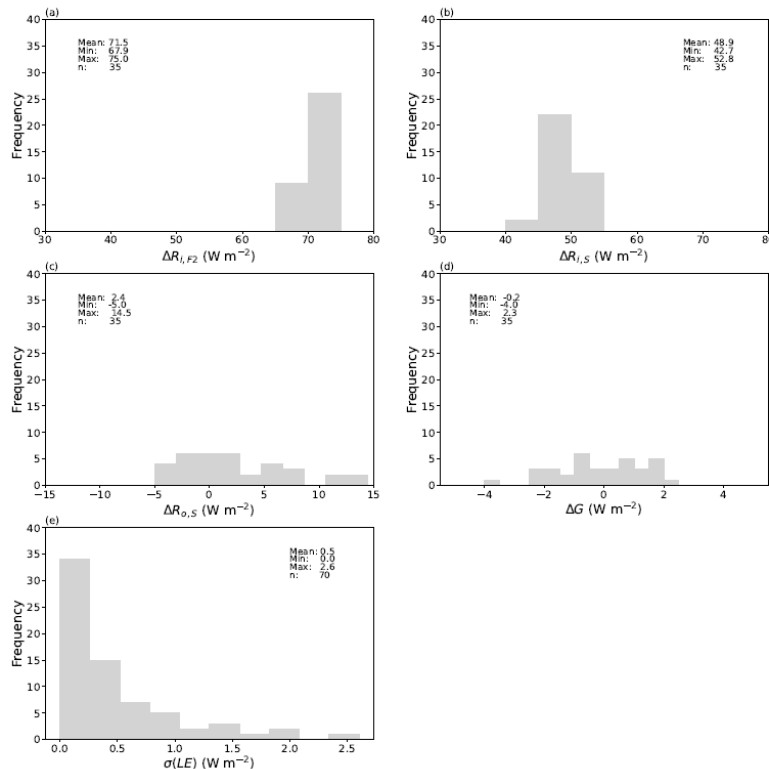

**Figure 17: Magnitude and uncertainty for the key experimental fluxes. Difference between the forced and ambient paired**
**experiments (n=35) in the incoming longwave radiation (a) at the top of the film ($\Delta R_{i,F2}$), (b) at the water surface ($\Delta R_{i,S}$), (c) outgoing longwave radiation from the water surface ($\Delta R_{o,S}$) and the (d) rate of enthalpy storage in the water bath ($\Delta G$). (e) Steady state standard deviation of the latent heat flux measurements $\sigma(LE)$ taken over all 70 evaporation experiments.**

The measured differences in those energy fluxes are shown in Fig. 17abcd. There is important variation in the radiative forcing
(Fig. 17ab) and the response (Fig. 17cd) between individual paired experiments. Despite that, we can obtain a useful overview of the accuracy of the measurements by examining the mean values for the radiative forcing and the response. The mean experimentally imposed longwave radiative forcing at the top of the film $\Delta R_{i,F2}$ is 71.5 W m$^{-2}$ (Fig. 17a) and at the water surface $\Delta R_{i,S}$ is 48.9 W m$^{-2}$ (Fig. 17b). The experimental uncertainty in a single measurement of $R_{i,S}$ was previously estimated as 2.2 W m$^{-2}$ (section 3.8). Assuming uncorrelated errors the uncertainty in the difference $\Delta R_{i,S}$ will be 3.1 W m$^{-2}$ (i.e., =
$\sqrt{(2.2^2 + 2.2^2)}$ ). To continue, the mean experimental radiative response at the water surface $\Delta R_{o,S}$ is 2.4 W m$^{-2}$. The error in a single measurement of $R_{o,S}$ was previously estimated as 2.9 W m$^{-2}$ (section 3.8) and using the same (uncorrelated error)

assumption the error in that difference will be 4.1 W m$^{-2}$. Hence in terms of the mean difference $\Delta R_{o,S}$ (= 2.4 ± 4.1 W m$^{-2}$) we have minimal change. Further, the mean value for the difference in enthalpy storage rate $\Delta G$ (i.e., variations in the departure from steady state) is smaller again at -0.2 W m$^{-2}$ which confirms that we have indeed experimentally achieved useful steady state conditions ($G \sim 0$, $\Delta G \sim 0$) across the entire experimental program. By comparison across all 70 evaporation experiments the uncertainty (±1sd) in the latent heat flux is up to 2.6 W m$^{-2}$ but in most (61 of 70) experiments, it is substantially less than 1 W m$^{-2}$ (Fig. 17e). Hence the difference $\Delta(LE)$ is likely to have an accuracy better than 2 W m$^{-2}$ in most paired experiments. That accuracy is more than sufficient to detect the evaporative response to a mean radiative forcing that averages 48.9 ± 3.1 W m$^{-2}$.

## 6 Discussion and Conclusions

The overall configuration of the wind tunnel was primarily governed by radiative considerations. The most important was to have a near transparent window through which we could admit different amounts of longwave radiation while independently controlling conditions inside the tunnel. The ideal design would have used a single layer of plastic film because that simplified the radiative transfer (cf. Fig. 6 v Fig. 7). However, in practice we found spontaneous condensation of liquid water onto the film interior often occurred at the highest wind speed (4 m s$^{-1}$) when using a single layer of film. The liquid condensate was clearly visible in the thermal imagery and we were unable to reliably measure the surface temperature of the water bath with the liquid condensate present. Instead, by using a double film layer we were able to experimentally eliminate the condensation but at the expense of creating a more complex radiative transfer problem.

A further challenge in determining the water surface temperature arose due to the moist air within the wind tunnel. We placed a small camera calibration spot within the view of the thermal camera and independently measured the temperature of that spot using a thermocouple. By that configuration our original conception was to compare the thermal camera and thermocouple measurements and apply that difference to the thermal camera measurement of the water surface to obtain the 'calibrated' water surface temperature. The failure of that conception led us to investigate the radiative transfer in more detail than we had originally anticipated. After further investigation the reason for the failure became evident – we had originally ignored the moist air corrections (Fig. 5). In particular, the temperature of the camera calibration spot is always very close to the air temperature in the tunnel and the moist air radiative correction is always very small irrespective of the ambient humidity in the tunnel. However, the water surface temperature was in the most extreme instance ~ 17°C colder (see T35-q7 in Fig. 15). More generally, the water surface was always colder than the tunnel air (Fig. 15). This requires a (non-negligible) moist air correction that will always be positive. Hence the original idea of transferring the camera calibration spot measurement to the water surface was found to be flawed and was abandoned. Instead we used a theoretical approach to model the underlying radiative transfer that proved successful (Fig. 9, Fig. C1b).

We found experimentally that the steady state temperature of the water bath closely approximated the theoretical wet bulb temperature. The theory we used to define the wet bulb temperature (Eqn 12) is based on concepts from classical *equilibrium* thermodynamics and the assumption of an adiabatic enclosure (Monteith and Unsworth, 2008). However, the wind tunnel experimental system described here is not an equilibrium system but instead operates at a steady state dis-equilibrium. The classical adiabatic saturation psychrometer also operates in a steady state dis-equilibrium and cools air by adding (liquid) water (Greenspan and Wexler, 1968). Here we have essentially reversed that operation by holding the properties (temperature, specific humidity) of the tunnel air constant and thereby cooling the shallow bath of liquid water down to a steady state temperature that closely approximates the theoretical 'equilibrium' wet bulb temperature. More detailed theory is readily available to analyse our steady state dis-equilibrium system (Greenspan and Wexler, 1968; Wylie, 1979; Monteith and Unsworth, 2008) but that is not necessary here since our aim was not to have a perfect wet bulb thermometer. Instead we note that the system is not strictly adiabatic because, by design, it allows longwave radiative exchange across the two film layers. That radiative exchange does not by itself invalidate the adiabatic assumption because there has to be a net absorption of heat by the air in the tunnel to violate the adiabatic constraint. However, we do anticipate small radiative modifications in the 300 mm high wind tunnel. A further consequence of the experimental configuration is that some (sensible) heat will also be conducted between the air in the tunnel and in the laboratory across the two film layers although we expect this to be minimal. Those two modes of heat exchange will ultimately depend on the difference in air temperature between the tunnel and the laboratory. That non-adiabatic exchange explains why we found consistent differences that varied with the laboratory air temperature (Fig. 13, Fig. 16).

In summary, the experimental system described here has been designed to investigate how evaporation is coupled to longwave radiation. In the traditional (Dalton-like) bulk formulae, evaporation is held to depend on the wind speed and the difference in specific humidity between the (near-saturated) surface and the ambient air. The traditional bulk formulae does not explicitly acknowledge any dependence on the longwave radiative fluxes. The experimental system can be used to hold the wind speed and specific humidity in the adjacent air at constant values while independently altering the incoming longwave radiation. By this design we are able to isolate any direct coupling of evaporation to the longwave radiative fluxes. In the paper we have shown that the steady state wind tunnel system provides reliable measurements and we can impose a controlled longwave radiative forcing of around 49 W m$^{-2}$ that is known to within $\pm 3.1$ W m$^{-2}$. When combined with a measurement accuracy of the evaporative response to that forcing that will be better than 2 W m$^{-2}$ we conclude that the new wind tunnel system is suitable for the experimental investigation of the coupling of evaporation to longwave radiation.

**Author Contribution**

MLR and CJS conceived the overall project and designed the experiments. AJR designed and constructed the wind tunnel. CJ carried out the experiments with assistance from MLR and AJR. MLR and CJS undertook the analysis. MLR prepared the manuscript with contributions from all co-authors.

**Data Availability**

The wind tunnel data is available at https://doi.org/10.5281/zenodo.8153246.

**Competing Interests**

The authors declare that they have no competing interests.

**Acknowledgements**

We thank Dr Chin Wong (Biology, ANU) for high level advice on the design, instrumentation and operation of the wind tunnel and Mr Peter Lanc (RSES, ANU) for developing the LabVIEW control software. The research was supported by the Australian Research Council (DP190100791). We acknowledge several helpful and insightful review comments by Dr Nathan Laxague that improved the article.

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

## Appendix A – Moist air absorptivity ($\beta_A$)

We used a PYTHON-based software package called Py4CAtS (Schreier et al., 2019) to solve the line-by-line radiative absorption over the wavenumber range 1-3000 cm$^{-1}$ at 243393 equally spaced wavenumbers. We calculated the moist air absorptivity of a slab of atmospheric air (total pressure of 1 bar) (Shakespeare and Roderick, 2021) at four different slab thicknesses ($z$; 0.01, 0.1, 0.3 0.5 m). In this calculation it was assumed that water vapour was the only radiatively active gas; including other less abundant greenhouse gases (e.g., $CO_2$) has negligible impact for the tunnel conditions (results not shown). We found that for a given slab thickness that the absorptivity primarily varied with the specific humidity with a small dependence on temperature over the range considered here (15, 25, 35, 45°C) (Fig. A1a). The dependence on slab thickness for these small thicknesses (i.e., close to zero) arose because many of the radiative absorption lines saturate rapidly as thickness increases from zero. Given the minimal sensitivity to temperature, we fitted an empirical power law to the moist air absorptivity as a function of specific humidity and slab thickness as follows,

$$\beta_A = 0.90\, z^{-0.68}\, q^{(0.44\, z^{-0.12})} \qquad (A1)$$

This empirical equation accurately described the moist air absorptivity over the thickness range considered here (Fig. A1b).

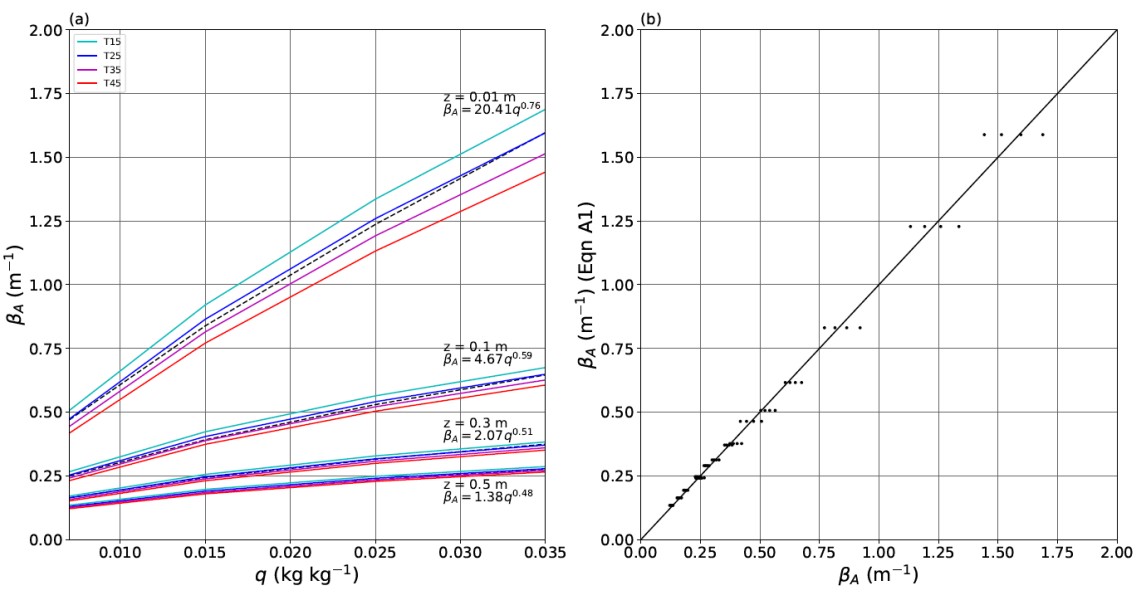

Figure A1: Dependence of moist air absorptivity on temperature, specific humidity and thickness of the moist air slab. (a) Moist air absorptivity as a function of specific humidity at different slab thicknesses ($z$ = 0.01, 0.1, 0.3, 0.5 m) and air temperatures ($T$ = 15, 25, 35, 45°C). Colours (see legend) indicate the temperature and the dashed lines show the indicated equation at each thickness. (b) Predicted moist air absorptivity using Eqn A1 compared with original data. Full line is 1:1. Linear regression is y = 0.997 x + 0.00, $R^2$ = 0.99, RMSE = 0.039, n = 64.

**Appendix B – Experimental determination of the bulk optical coefficients of the film**

To determine the bulk optical properties of the plastic film we carried out a series of separate experiments by generating a known longwave radiative (i.e., black body) flux and measuring the transmission of that flux through 1 and/or 2 layers of film.

The configuration is shown in Fig. B1. We connected a constant temperature water bath ([4] in Fig. B1) via a circulatory system to a heat exchanger ([3] in Fig. B1) on which we sat a painted copper slab (12.5 mm thick, emissivity of paint = 1, [2] in Fig. B1). Heat was rapidly conducted from the heat exchanger into the copper slab whose temperature was continually monitored using the laboratory temperature reference probe ([5] in Fig. B1) inserted into the middle of the copper slab via a drilled hole. By changing the temperature of the copper slab in five set steps (10, 20, 30, 40, 50 °C) we could generate a known

(assumed isotropic) longwave radiative flux that then travelled through the moist air and film (either 1 or 2 layers) to the thermal camera ([1] in Fig. B1).

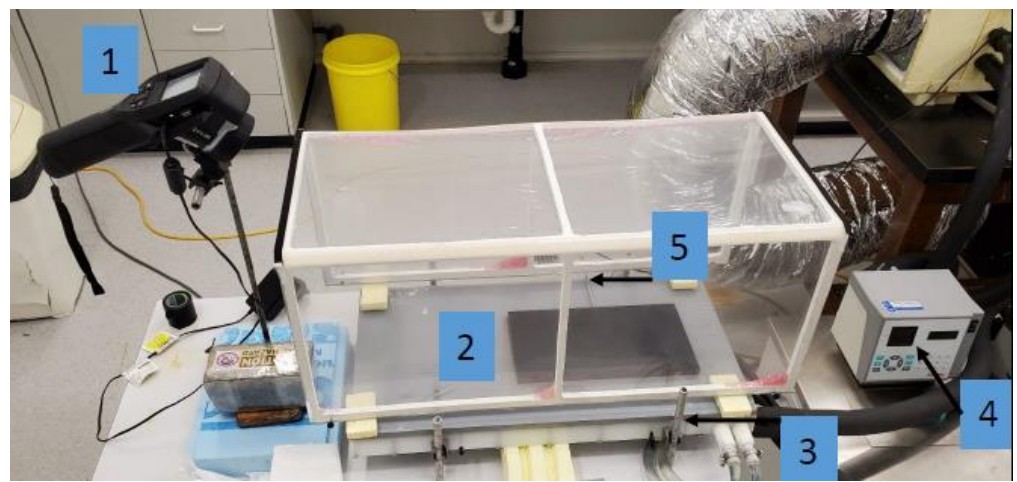

**Figure B1: Experimental configuration for estimating bulk optical properties of the plastic film. Key numbers as follows:**
**[1] Thermal camera (FLIR: Model E50); [2] Copper plate (painted black); [3] Heat exchanger connected to a [4] constant temperature water bath (Julabo: Model PP50); [5] temperature probe (HART Scientific: Model 1521).**

To estimate the bulk transmission through the film we used the above configuration (Fig. B1) with a single layer of film (see theory in Fig. 6). We measured the outgoing longwave radiation arriving at the thermal camera ($R_{O,C}$) through 1 film layer at

five different copper plate temperatures ($T_0$; 10, 20, 30, 40, 50 °C) and at two different laboratory temperatures ($T_L$; 19, 31 °C) giving a total of 10 observations. By inspection of Fig. 6, we relate the radiative flux ($= R_{O,C} - dR_{0-1} - dR_{1-2}$) to the experimentally varied temperatures ($T_0$, $T_L$) and bulk optical properties using,

$$R_{O,C} - dR_{0-1} - dR_{1-2} = \tau\sigma T_0^4 + (\alpha + \beta)\sigma T_L^4 \qquad , \tag{B1}$$

with the moist air corrections calculated at the prevailing specific humidity ($q_L = 0.005$ kg kg$^{-1}$) using
$dR_{0-1}(T_0, T_L, q_L, 0.44)$ and $dR_{1-2}(T_0, T_L, q_L, 0.14)$. Note that the relevant distance for the moist air corrections used here is

along the path to the camera. We further note that by this experimental configuration, we cannot distinguish the reflection from the absorption (Eqn B1) and we used this approach to determine their sum. The least squares solution for the bulk optical parameters using the 10 available observations was (Fig. B2),

$$\tau = 0.908 \mp 0.029\ (\mp 1sd), \qquad (\alpha + \beta) = 0.092 \mp 0.032\ (\mp 1sd) \qquad , \qquad \text{(B2)}$$

with an overall RMSE of 2.0 W m$^{-2}$. The experimental results were in accord with theoretical expectations (Eqn 2) with the sum of the transmission and the reflection plus absorption equal to 1 within experimental uncertainty. The results show the plastic film was highly transmissive with some 90.8% of the incident longwave radiation transmitted.

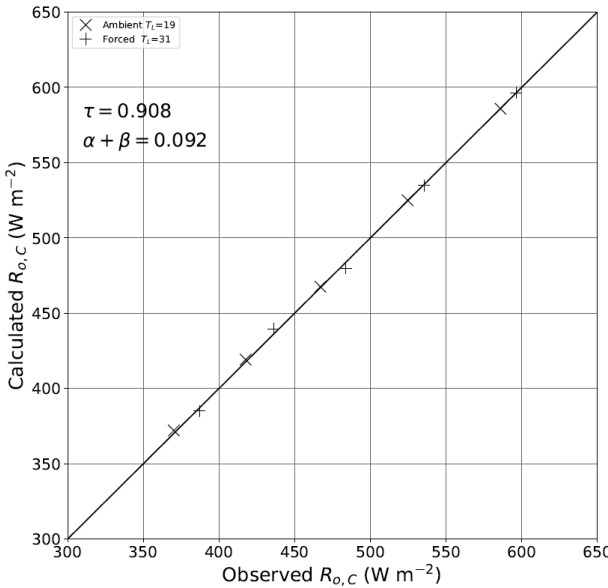

**Figure B2: Experimental estimate of bulk transmission coefficient of the film. Plot shows calculated versus observed longwave radiation arriving at the thermal camera using least squares estimates for the bulk coefficients ($\tau = 0.908$, $\alpha+\beta = 0.092$) (Linear regression: y = 0.9941 x + 2.9, $R^2 > 0.999$, n = 10, RMSE = 2.0 W m$^{-2}$). Full line is 1:1.**

One way to separate the reflection from the absorption of the film was to independently vary the temperature of the film relative
to that of the surrounding air thereby altering the emitted component of the radiative flux. After many trials we eventually adopted an approach that used two films mounted onto the PVC frame (with the same 10 mm air gap) along with the copper plate (Fig. B1). To alter the temperature of one of the films, we located an air heater (air curtain) slightly below the lower film and passed air of a fixed temperature along the film. In reality this approach would have also changed the temperature of a thin slab of moist air below the lower film but that complication was ignored. For the experiment we measured the outgoing
longwave radiation arriving at the thermal camera ($R_{O,C}$) through 2 film layers at five different copper plate temperatures ($T_0$; 10, 20, 30, 40, 50°C) while changing the temperature of the lower film in three steps ($T_1$; 25, 35, 45°C). The experiment was

conducted at a single laboratory temperature ($T_L$; 19°C) giving a total of 15 observations. By inspection of Fig. 7, the relevant equation for the outgoing longwave flux arriving at the thermal camera ($R_{O,C}$) under the stated conditions is,

$$R_{O,C} = \tau^2 \sigma T_0^4 + dR_{0-1} + dR_{1-2} + dR_{2-3} + \tau\beta\sigma T_1^4 + (\beta + \alpha\tau^2 + \alpha)\sigma T_L^4 \qquad , \qquad (B3)$$

with the moist air corrections calculated at the prevailing specific humidity ($q_L = 0.005$ kg kg$^{-1}$) using $dR_{0-1}(T_0, T_L, q_L, 0.44)$, $dR_{1-2}(T_0, T_L, q_L, 0.015)$ and $dR_{2-3}(T_0, T_L, q_L, 0.125)$. To estimate $\alpha$ we first set $\tau = 0.908$ (Eqn B2) and varied $\alpha$ over the permissible range (0 to 0.092) subject to the constraint that $\alpha+\beta = 0.092$ (per Eqn 4). At each trial value of $\alpha$ (and hence $\beta$) we compared the predicted and observed outgoing longwave flux at the camera using the 15 available observations and calculated the RMSE. The result showed a clear minimum (Fig. B3a) with the best fit value for $\alpha = 0.047$ (and hence $\beta = 0.045$) with an

overall RMSE of 3.4 W m$^{-2}$ (Fig. B3).

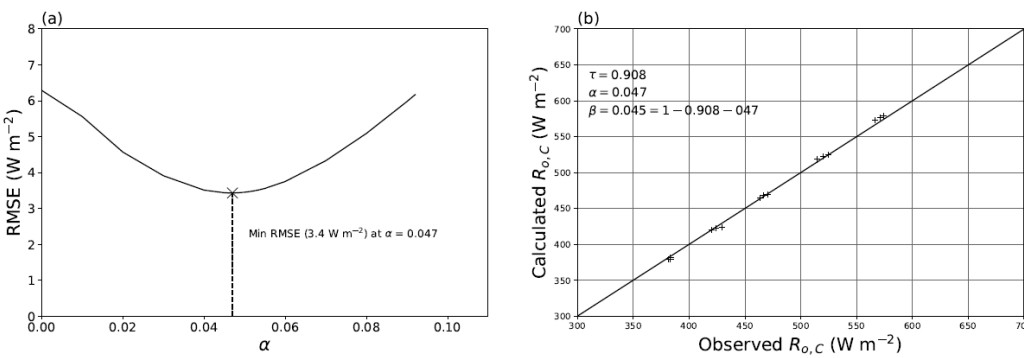

**Figure B3: Experimental estimate of the bulk reflection ($\alpha$) and absorption ($\beta$) coefficients. (a) RMSE (Eqn B3) as a function**
**of $\alpha$, and the (b) calculated versus observed longwave radiation arriving at the thermal camera based on the bulk optical properties (Linear regression: y = 1.0445 x – 20.5, $R^2 > 0.999$, n = 15, RMSE = 3.4 W m$^{-2}$). Full line is 1:1.**

We note that the latter experiment to estimate the reflection and absorption coefficients (Fig. B3b, RMSE: 3.4 W m$^{-2}$) was not as precise as the former experiment to estimate the transmission coefficient (Fig. B2, RMSE: 2.0 W m$^{-2}$). Inspection of the

prevailing equations (Eqn B3) shows that the radiative transfer is much more sensitive to errors in the transmission compared to the reflection and/or absorption. With that we note that the most useful estimate of the error is the ultimate experimental error when estimating the incoming longwave radiation at the water surface using the complete theory. As we show in the main text, with some very minor adjustments to the parameter values we were able to estimate the incoming longwave radiation at the water surface with an RMSE of 2.2 W m$^{-2}$ (Fig. 9b) that was very similar to the error found when estimating the bulk

transmission (Fig. B2, RMSE: 2.0 W m$^{-2}$). This was anticipated since as noted above, the bulk transmission coefficient is the most important of the three optical variables.

## Appendix C – Estimating the geometric parameter $g_1$

During the evaporation experiments we simultaneously recorded the longwave radiation arriving at the thermal camera from the water surface (of as yet unknown temperature $T_S$) and from the calibration spot whose temperature was also measured independently using a thermocouple ($T_T$, Fig. 2b). Hence we developed a semi-empirical equation using the available calibration spot observations embedded in the evaporation experimental data ($n = 70$) to experimentally determine the required geometric parameter ($g_1$). By inspection of Fig. 7 and Fig. 8, the outgoing longwave radiation arriving at the thermal camera from the calibration spot ($R_{o,C,T}$) is written as,

$$R_{o,C,T} = g_1 \left(\sigma T_L^4\right) + (1 - g_1)\left(\tau^2 \sigma T_T^4 + \tau\beta\sigma T_A^4 + (\beta + \alpha\tau^2 + \alpha)\sigma T_L^4 + dR_{0-1} + dR_{1-2} + dR_{2-3}\right) \qquad , \qquad \text{(C1a)}$$

with $g_1$ an (as yet) unknown geometric parameter that is a direct analogue of $g_0$. The moist air corrections given here are calculated using $dR_{0-1}(T_T, T_A, q_A, 0.44)$, $dR_{1-2}(T_T, T_A, q_A, 0.015)$ and $dR_{2-3}(T_T, T_L, q_L, 0.125)$. Note that the first two moist air corrections ($dR_{0-1}$, $dR_{1-2}$) are negligible since $T_A$ and $T_T$ are almost equal.

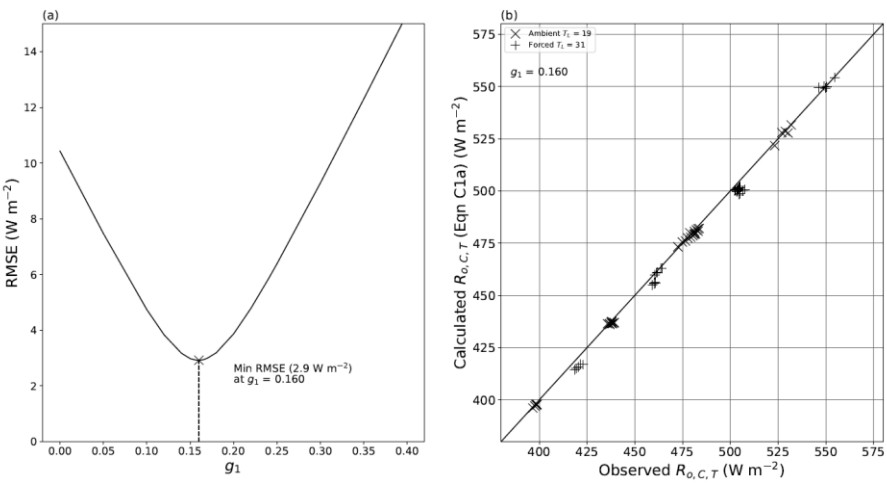

**Figure C1: Experimental estimate of the geometric parameter $g_1$. (a) RMSE (Eqn C1a) as a function of $g_1$ highlighting the identified minimum value. (b) Comparison of observed and calculated outgoing longwave radiation arriving at the thermal camera from the calibration spot using the optimal value for $g_1$ (= 0.160) (Linear regression: y = 1.002 x – 2.9, $R^2$ = 0.997, RMSE = 2.9 W m$^{-2}$, n = 70). Full line is 1:1.**

We determined $g_1$ by selecting the value with a minimum RMSE ($g_1 = 0.160$, Fig. C1a) and with that numerical value, Eqn C1a becomes,

$$R_{o,C,T} = 0.160 \left(\sigma T_L^4\right) + 0.840 \left(0.8281\, \sigma T_T^4 + 0.0364\, \sigma T_A^4 + 0.1314\, \sigma T_L^4 + dR_{0-1} + dR_{1-2} + dR_{2-3}\right) \qquad , \qquad \text{(C1b)}$$

and has been used to predict the outgoing longwave radiation arriving at the thermal camera from the calibration spot (Fig. C1b). Again we note that the thermal radiation arriving at the camera from the camera spot is predominantly determined by

the black body emission from the spot but is also impacted by variations in $T_L$ with a very small contribution from $T_A$. The results showed a tight fit (RMSE = 2.9 W m$^{-2}$, Fig. C1b) with no obvious bias under either ambient ($T_L$ = 19°C) or forced ($T_L$ = 31°C) conditions.


**Appendix D – The psychrometer constant (γ) as a function of air temperature and relative humidity**

The psychrometer constant γ (Pa K$^{-1}$) given by,

$$\gamma = \frac{P\,c_P}{\varepsilon\,L} \qquad , \qquad\qquad\qquad (D1)$$

with $P$ the total air pressure, $c_P$ the specific heat of air, $\varepsilon$ the ratio of the molecular mass of water to air (~ 0.622) and $L$ the latent heat of vaporisation (Monteith and Unsworth, 2008). In many practical applications the specific heat is often taken as
that for dry air but the formal theory requires the integrals to be taken over the actual (moist) air (Monteith and Unsworth, 2008; Greenspan and Wexler, 1968). With the specific heat for moist air slightly larger than for dry air and $L$ declining slightly with temperature, the numerical value for γ varies slightly with temperature and relative humidity. At a total pressure of 1 bar γ is 66 Pa K$^{-1}$ at 15°C (Fig. D1) with minimal changes due to variation in relative humidity. At 45°C in completely dry air γ is 68 Pa K$^{-1}$ but increases to 71 Pa K$^{-1}$ in completely saturated air (Fig. D1). The results presented in the main text are not
especially sensitive to the numerical value and we use a constant value for γ (= 68 Pa K$^{-1}$) for all calculations in this paper.


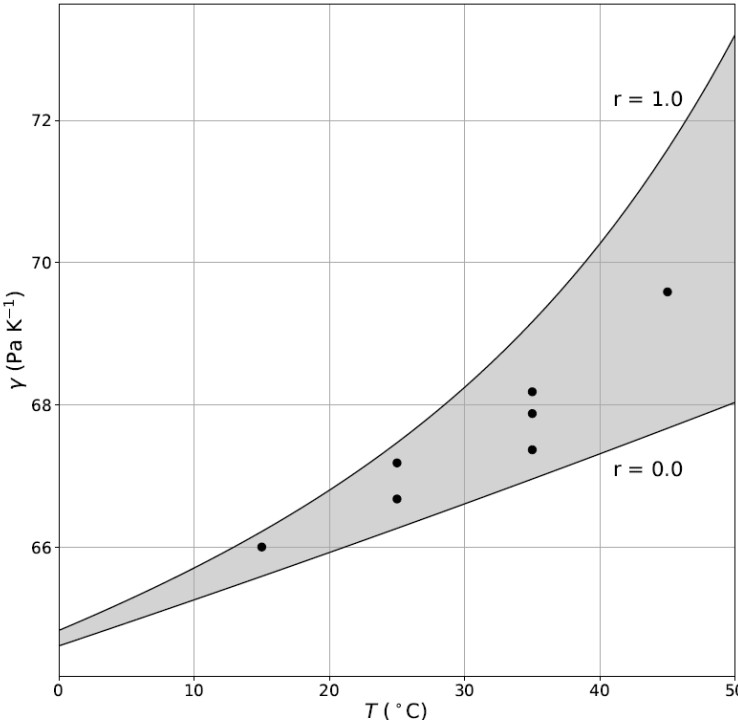

**Figure D1: The psychrometer constant (γ) as a function of air temperature and relative humidity (*r*) at total air pressure of 1 bar. The shaded area denotes the bounds between dry (*r* = 0.0) and saturated (*r* = 1.0) moist air. The dots depict the seven temperature-humidity combinations used in the experiment (Fig. 3). Data for specific heat and latent heat of vaporisation are from the International Association for the Properties of Water and Steam (IAPWS) database (Wagner and Pruß, 2002).**