# Peer review of "Evaluation of a wind tunnel designed to investigate the response of evaporation to changes in the incoming longwave radiation at a water surface"

_EGUsphere, 2022_

## Community Comment (CC1)

**Response to Reviewer Comment 1 (https://doi.org/10.5194/egusphere-2022-986-RC1)**

**Original review in ITALICS. Author Response in bold.**

The authors intend to investigate the influence of longwave radiation on evaporation. They claim that because longwave radiation is absorbed in the top 20  $\mu$ m at the ocean surface this heat source/sink might cause significant deviations of the surface temperature from the underlying water. This effect changes the saturation water vapor pressure at the water surface. Therefore the concentration difference between the water surface and a reference height is influenced and with it the water vapor flux and evaporation rate. They claim that this potentially large effect is completely ignored in bulk formulas for evaporation.

The authors therefore want to study the effect of longwave radiation on evaporation in a wind tunnel specifically designed for this purpose. In the first paper (egusphere-2022-986) they focus on a thermodynamic characterization of the facility, and in a second one (egusphere-2022-986) on the radiative characterization.

The subject of the study is ill-defined and the described facility and instrumentation not really suitable for the intended purpose. Therefore the reviewer recommends rejection of the publication of the two manuscripts.

We thank the reviewer for the time taken to read the manuscript.

The review is clear-cut (e.g., ".. ill-defined ..", ".. not really suitable for the intended purpose ..", ".. rejection ..").

The very brief review does not include an assessment of any of the manuscript sections (2 Design and Operation, 3 Thermodynamic Evaluation, etc.).

Instead the review is based on an assertion that we are not aware of previous work. The (implied) implication is that the basic idea is fatally flawed.

The assertion/s are wrong.

**Instead, we are conducting a series of very well-defined laboratory experiments as described below.**

The claim that "mass transfer formulations for evaporation ... not directly consider the langwave radiative fluxes" is simply not correct. The reviewer did not perform a systematic literature search, but quickly found two almost 30 years old papers, dealing with the subject: McPhaden (https://doi.org/10.1175/1520-Zhang and 1995 0442(1995)008<0589:TRBSST>2.0.CO;2 ) and Fairall et al. 1996 (https://doi.org/10.1029/95JC03190). The actual version 3.6 of the COARE algorithm

published on zenodo explicity includes longwave irradiation (named there IR flux): Bariteau et al., 2021 (https://doi.org/10.5281/zenodo.5110991)

We are correct.

To see that, let us examine the first work cited by the reviewer (Zhang and McPhaden 1995). In that work the calculation of the sensible (their Eqn 1a) and latent (their Eqn 1b) heat fluxes are reproduced from their paper as follows:

$$F_h = C_p \rho C_h U(T_s - T)$$
(1a)

$$F_q = L_v \rho C_e U[q^*(T_s) - q]$$
(1b)

With reference to (their) Eqn 1b, this is the classical Dalton-type bulk formula for evaporation that we referred to as the mass transfer formulation in the manuscript. Neither the incoming or outgoing longwave radiation is explicit in the equation. (Note: The cited Fairall et al 1996 and associated COARE reference cited by the reviewer use exactly the same mass transfer formulation (see Eqn 17 in Fairall et al. 2003, J Climate 16: 571-591)). What the cited references do is calculate the evaporation using the mass transfer formulation (e.g., Eqn 1b above). They then combine the latent and sensible heat fluxes with the radiative fluxes to define the surface energy balance. We have done this ourselves on many occasions and this can be readily confirmed by a literature search. We note that this approach is described in standard texts (see references we cited on line 25 and line 40 of the manuscript) which present the same method as in the references cited by the reviewer.

However, we are going well beyond this long-used mass transfer formulation.

Instead, the fundamental basis for our new laboratory-based facility is described in lines 74-76 of the manuscript as follows:

"The unique feature is an augmented capability to independently vary the incoming longwave radiation at the water surface whilst holding the other variables fixed. The scientific rationale of this approach was to isolate the effect of a change in the incoming longwave radiation on both evaporation and surface temperature."

We can explain this statement with reference to Eqn 1b (above). With  $L_v$  (the latent heat of vaporisation) and  $C_e$  (transfer coefficient) both more or less constant, one can envisage an experimental configuration where  $\rho$ , U,  $q^*(T_s)$  and q in Eqn 1b were all held fixed. Eqn 1b would predict no change in the latent

heat flux if the incoming longwave radiation was independently varied because the incoming longwave radiation is not directly represented in Eqn 1b.

In our experiments we hold  $\rho$  (air density), U (wind), q (specific humidity of air) and T (temperature of air) constant. We then vary the incoming longwave radiation and measure how the surface temperature ( $T_s$ ) and latent heat flux respond. Hence our work is not "*ill-defined*". Instead we are conducting a series of well-defined laboratory experiments to experimentally examine the fundamental basis of the mass transfer approach itself. To our knowledge our work represents the first-ever experiment to examine the fundamental basis of the mass transfer approach. For this reason it is critical to fully document the technique/s - hence our submission to the AMT journal which seemed ideal for this purpose.

The authors are obviously not familiar with the extended research work on the difference between the ocean surface temperature and the underlying bulk water ("cool skin"). Much of the pioneering work was done by Katsaros, see, e. g., Katsaros 1980 (https://doi.org/10.1007/BF00117914) or Katsaros 1990 (https://doi.org/10.1007/978-94-009-0627-3\_9). A comprehensive account of the near-surface layer of the ocean is given in the monograph of Soloviev and Lukas 2014 (https://doi.org/10.1007/978-94-007-7621-0).

In response we confirm that we are aware of the extended body of work (especially in remote sensing) on differences between the bulk and skin temperature over the ocean (and over lakes). However, again we re-iterate, we are examining a more fundamental question: the validity of the mass transfer formulation itself.

There are still, of course, open question. Most of them are related to the mechanisms of the transport from the ocean surface down to the bulk water, especially the influence of wind waves. The wind tunnel built by the authors is not suitable to address these questions because of the tiny and shallow water basin. A large wind-wave facility, such as the LASIF at the University of Marseilles (France) would be required for such studies (https://www.osupytheas.fr/?-LASIF-Grande-Soufflerie-air-eau-de-Luminy-&lang=en) and instrumentation and methods to image the water surface temperatures and temperature profiles in the aqueous viscous boundary layer.

We agree with the reviewer that there is much science still to be done on many important topics.

We also agree with the reviewer that the facility we have built is not suitable to address the influence of wind waves on heat transfer into the interior of the ocean. The reason is that this is not the scientific question we are addressing.

The question we address: the validity of the mass transfer formulation for evaporation that has been in widespread use for the last 220 years, is, in our

opinion at least, a very important scientific topic. In that context, the experimental facility we have established represents the first-ever examination of the validity of the (220 year old) mass transfer formulation for evaporation.

Michael L. Roderick & Callum J. Shakespeare (on behalf of all authors), 15/3/2023

Postscript (editorial question):

We submitted two manuscripts to the AMT journal (titled Parts 1 and 2 in September 2022). This is the first review of that work we have received (in March 2023) and the review has recommended rejection of both manuscripts. However, only Part 1 has appeared as a preprint on the journal website. We seek guidance from the editor on the status of the Part 2 manuscript. We have previously been informed that Part 2 was suitable for the journal and that an editor has been assigned but we have not received a preprint and the Part 2 manuscript is not posted on the journal website.

---

## Author Comment (AC1)

**Response to Reviewer Comment 2**
(https://doi.org/10.5194/egusphere-2022-986-RC2)

*Original review in ITALICS.*
**Author Response in bold.**

*A review of "Evaluation of a wind tunnel designed to investigate the response of evaporation to changes in the incoming longwave radiation at a water surface. I. Thermodynamic characteristics" by Michael L. Roderick, Chathuranga Jayarathne, Angus J. Rummery, and Callum J. Shakespeare.*

*In this manuscript, the authors describe an experimental setup for isolating the effect of longwave radiative flux on evaporation from a water surface. In short, the setup includes a wind tunnel containing a small water bath, with embedded sensors for measuring the humidity of the air and the temperature of the air and water. Additionally, incoming longwave radiative flux was measured via pyrgeometer and the water skin temperature was measured via microbolometer. The room containing the setup was described as being "temperature-controlled", though the cooling/ventilation system operated beyond the control of the authors, producing a noticeable oscillation in ambient temperature and humidity.*

*The topic as described by the authors is certainly of interest to the readership of AMT, and the manuscript was written with clear language. However, I have major concerns with some core elements of the laboratory setup. Furthermore, I found it difficult to assess the importance of this manuscript as an independent piece of research. The whole project is motivated by a desire to investigate the specific impact of longwave radiative flux on evaporation, but the details of the radiative component are left to the (as of yet unseen) part 2. I don't know that part 1 stands on its own as a meaningful contribution- is the result that bulk water temperature is sometimes close to the wet bulb temperature of the air above it? In any case, I believe that the work the authors have done can contribute the the body of knowledge, but I strongly recommend that they significantly revise this manuscript. Without seeing the second manuscript, I can only recommend that the revision to part 1 should include more details regarding the radiometric measurements (and results related to the total heat budget calculations). It may be that such a revision would combine the two parts- or that new radiative measurements need to be made with higher quality instrumentation, but it's impossible to say having only seen part 1 of the work.*

**We thank the reviewer for the careful and helpful review.**

**Both the second and third major comments made by the reviewer were good points that related directly to the radiative measurements that were the subject of a second (radiative) manuscript. We were able to adequately address the reviewers concerns using results taken directly from the second (radiative) manuscript.**

The review comments raise a more general point: why not combine the manuscripts? For relevant background, we submitted two manuscripts to the AMT journal (titled Parts 1 and 2 in September 2022). However, only Part 1 (Thermodynamics, egusphere-2022-986) has appeared as a preprint on the journal website. For the Part 2 (Radiative, egusphere-2022-988) manuscript, an associate editor has been assigned but we have not received a preprint and it has not yet been sent for any form of review as far as we are aware. We note that the associate editors for the two articles are different. We had anticipated that both articles would be handled by a single associate editor and sent simultaneously to the same reviewer/s. This would have been helpful in this case since the 2 of 3 major concerns by the reviewer related to content in the second manuscript.

In response we will independently contact the editorial staff to discuss options to combine both manuscripts into a single manuscript.

*Major Concerns*

*The room that was described as "temperature-controlled" showed a regular oscillation of ambient air temperature and specific humidity (approximately 1000 second period). This has a meaningful impact on what should be a sensitive measurement. To this point, the oscillation in incoming longwave radiative flux appears to track (in both shape and phase) the oscillation in humidity- not temperature. Does this mean that radiative flux sensed by the pyrgeometer is representative of more volumetric (path) absorption/emission in conditions of elevated humidity? In any case- unless this periodic behavior can be leveraged as an asset in the heat flux balance calculations (i.e., the longwave radiative flux is the only oscillating flux, allowing the authors to parse its effect on the evaporation), I believe it will be a major liability in the calculations.*

Close examination shows that the oscillation in longwave ($R_{i,F2}$ in Fig. 5e) due to the oscillation in the laboratory temperature control closely tracks the laboratory air temperature ($T_L$ in Fig. 5a, and equivalent to Black body at $T_L$ in Fig. 5e) and not the specific humidity.  Hence the variations in specific humidity (e.g. $q_L$ in Fig. 5c) have no measurable impact on the longwave radiation.

The variations in specific humidity do follow the variations in laboratory air temperature but we know that the specific humidity has little impact on the radiative flux. We know that because at any given instant of time, the water vapour is more or less at the same temperature as the bulk air and the walls of the room (= $T_L$). Some of the (near blackbody) radiative flux emitted by the walls at $T_L$ would be absorbed by water vapour (also at $T_L$). The effective optical path length will change with the specific humidity as noted by the reviewer and the absorption will increase. But the absorption and re-emission by the water

vapour occurs at the same temperature as the air and the walls of the room and this does not materially alter the flux. There is an in-depth explanation of the physics in the second radiative manuscript. With that, we note that the temperature-controlled room behaves as a black body, all be it, with an oscillation induced by the heating system.

We handled the oscillation by taking the steady state period to be integer multiple of the oscillation period (as described in lines 286-288). As a consequence the standard deviations we reported (Figs 7a, 7c, 8d) are overestimates. This is discussed in detail in the manuscript (lines 433-450). We can extend that discussion here by using results from the second radiative paper which show an overall uncertainty of around 2 to 3 W m$^{-2}$ in estimating the incoming and outgoing longwave fluxes at the water surface. Hence any residual effect of the oscillatory behaviour on the evaporation or longwave forcing is small relative the imposed longwave forcing at the water surface of 50 W m$^{-2}$.

*Since the thin film-covered window only occupies a small segment of the hemisphere above the pool, how will the authors account for the difference between radiative flux emanating from the walls of the room and the flux emanating from the inside of the wind tunnel? Would a view factor (or some sort of solid angle accommodation) be helpful in accounting for this difference? Regardless, I'm not certain that a hemispheric pyrgeometer is the ideal sensor for this type of indoor, spatially heterogeneous measurement.*

This is a perceptive comment. In response that is exactly what we did. We show below Fig. 10 from the second (radiative) paper.

[Figure]

Figure 10: Schematic drawing showing separate contributions to the incoming longwave radiation at the water surface. The diagram is a cross section along the centreline of tunnel showing the hemispherical geometry used to estimate the incoming longwave radiation at the water surface arriving from the tunnel, film and PVC frame.

To model the incoming longwave flux at the base of the tunnel we developed a theoretical model (see Fig 10 above, taken from the second radiative paper) that separately accounted for the longwave fluxes from the inside of the tunnel and from the room. The theoretical model was then evaluated using hemispherical measurements made using the radiometer located at the same position as the water bath. The theoretical model was found to predict the incoming longwave at the water surface with an RMSE of 2.2 W m$^{-2}$ (see Fig. 11b below, also taken from the radiative manuscript).

[Figure]

Figure 11: Comparison of theoretical and observed incoming longwave radiation at the water surface. (a) Uses g0 = 0.156 as per theory (Linear regression: y = 0.9855 x + 7.3, R2 = 0.991, RMSE = 3.1 W m-2, n=20). (b) Tuned to locate the value of g0 (=0.128) with the best fit (Linear regression: y = 0.9772 x + 9.7, R2 = 0.997, RMSE = 2.2 W m-2, n=20). Full lines are 1:1.

*I mentioned the need for more information about the radiometry. It strikes me that measurements of the skin temperature are missing here. But even if they were included- can the authors make a case that the FLIR E50 is up to the challenge of providing high-quality radiometric measurements? It appears to be a handheld system that is optimized for qualitative evaluation of heat sources in industrial/construction use cases. If the authors are able to provide the results of blackbody calibration to establish the instrument's accuracy, stability, and low noise, that might put these fears to rest. But non research-grade microbolometers are notorious for being poor in those categories and are exceptionally prone to drift (which might be the worst type of error one could have when making the "steady state dis-equilibrium" measurements described here). A cooled single-point infrared radiation thermometer is usually regarded as the superior instrument for these sensitive measurements.*

**The FLIR E-50 model has a nominal temperature resolution of 0.05 degC. To test the accuracy, precision and overall repeatability, we constructed a black body (a copper plate covered with black paint of emissivity 1) and by varying the temperature of the plate we could compare the flux measured with the FLIR E-50 with that measured by the Kipp and Zonen CNR1 net radiometer under a variety of conditions. We found the FLIR E-50 measurements to be repeatable and the recorded flux was the same (within error tolerances) as that of the research grade Kipp and Zonen radiometer.**

**As noted by the reviewer, the surface temperature measurements were missing here because they are included in the second radiative manuscript.**

*Minor Comments*

- *For most parameters, variability is represented via standard deviation (or 95% confidence intervals). However, several quantities (ambient air temperature, ambient air humidity, incoming longwave radiative flux) oscillate with the room's cooling system. Perhaps the authors could report the amplitude of oscillation for these quantities?*

**Good point. For the variables mentioned (air temperature ($T_L$) and specific humidity ($q_L$) in the laboratory, incoming longwave flux at the top of the film ($R_{i,F2}$)) we have reported the standard deviation ($\sigma$) as the reviewer noted. The oscillation was the main source of (temporal) variation (see lines 287-288). With that, we note that the standard formula (for a sine wave, $Amplitude = \sqrt{2}\,\sigma$) can be used to estimate the amplitude.**

- *The variation in enthalpy (Figure 9) is computed via temperature difference between the beginning and end of steady-state period. What could be learned from performing multiple short-window linear least-squares regressions during the steady state period, thereby obtaining a LHF estimate for each subwindow?*

**Good point. We tried this during our initial experiments and found the "noise" became too great over short time intervals in the LE (what the reviewer called LHF, latent heat flux) and also in the enthalpy change in the water bath (what we called *G*). Hence we were obliged to calculate the steady state values of the fluxes using longer periods that ranged from 850 (i.e., ~ 14 minutes) to 3300 (i.e., 55 minutes) seconds as described on line 442 of manuscript.**

- *The Kipp & Zonen radiometer is said to be located 'in the laboratory (but outside the tunnel)' during evaporation experiments. Could this be pointed out in Figure 2?*

**During routine evaporation experiments the Kipp and Zonen radiometer was located inside the cardboard box sitting on the top of the wind tunnel (to the left of the number 3 in Fig. 2 as shown below). This was done to avoid thermal interference. For example, if the radiometer was sitting on top of the tunnel in the open, it would respond to the body heat of staff members as they walked past. Shielding the radiometer by placing it inside the cardboard box proved a simple solution that removed thermal interference.**

[Figure]

90  Figure 2: Photograph of the wind tunnel in the temperature controlled room of the Geophysical Fluid Dynamics Laboratory. Key numbers as follows: [1] Water bath and digital balance (AND Corporation: Model GX-6100); [2] Variable speed fan; [3] Thermal camera (FLIR: Model E50); [4] Camera Spot (used for thermal camera calibration); [5] Radiator (for air temperature control); [6] Constant temperature water bath (Julabo: Model PP50); [7] Humidity/Temperature sensor (for measuring tunnel air, VAISALA: Model HMP140); [8] Humidity/Temperature sensor (for measuring laboratory air, VAISALA: Model HMP140); [9] Temperature
95  sensor (thermistor for measuring tunnel air, Thermometrics NTC: Model FP07DA103N ); [10] Vapour source (humidifier for humidity control of tunnel air); [11] Digital controller.

- *It is a bit taxing to jump between Figure 3 and the body of the manuscript to find the definitions for the state variables. I recommend adding descriptive labels on the figure or more content to the caption.*

**We can do that is a revised version of the manuscript.**

- *For Figure 3, I think that a hashmark or dot pattern along the solid regions of the tunnel setup would aid the reader in interpreting the setup. Furthermore, arrows or streamlines in the tunnel portion would be helpful.*

**We can do that is a revised version of the manuscript.**

- *Figure 11 is effective, but the extra information presented in Figure 12 is difficult to digest. Perhaps the authors could establish the concept in Figure 11 (as already done), then replacing Figure 12 with scatter plots that relate Tw, TB (or better, Tskin), and the inferred psychrometer constants.*

**This is a problem caused by having two manuscript parts that describe the (i) thermodynamics and (ii) radiation separately.**

**We have actually done what the reviewer suggested in the second radiative manuscript. We could not do that in the thermodynamic manuscript because the measurement of the surface "skin" temperature had not yet been described. We show Fig. 13 from the second radiative at manuscript below: with $T_B$ (x-axis) the mean bulk water temperature, and $T_S$ (y-axis) the measured surface temperature.**

[Figure]

Figure 13: Comparison of observed bulk water temperature ($T_B$) with calculated surface temperature of the evaporating water bath ($T_S$) during all evaporation experiments (n = 90). Full line is 1:1. Linear regressions for ambient (blue dashed line, y = 1.155 x – 2.97, $R^2$ = 0.998, RMSE = 1.3°C, n = 45) and forced (red dashed line, y = 1.139 x – 4.32, $R^2$ = 0.998, RMSE = 1.3°C, n = 45) conditions also shown.

**Michael L. Roderick & Callum J. Shakespeare (on behalf of all authors), 24/3/2023**

---

## Author Response (AR1)

**Overall Response**

**We originally submitted two separate manuscripts (Part 1, thermodynamic, egusphere-2022-986; Part 2, radiative, egusphere-2022-988) to describe our experimental facility for investigating the coupling of longwave radiation and evaporation.**

**We have followed the editorial recommendation to combine both manuscripts into a single paper. The title for the new combined manuscript is: "Evaluation of a wind tunnel designed to investigate the response of evaporation to changes in the incoming longwave radiation at a water surface".**

**The original two part manuscript included in Part 1, 7750 words+12 Figures and Part 2 included 7200 words + 14 Figures + 1 Table. The new single part manuscript includes 12000 words (main text) + 2 Tables + 17 Figures + 4 Appendices (that include 6 figures and 1400 words).**

**We believe that the new combined manuscript has greatly improved the accessibility and readability of the research.**

**Our responses to the original reviews are shown below.**

*Original review in ITALICS.*
**Author Response in bold.**

**Response to Reviewer Comment 1**
(https://doi.org/10.5194/egusphere-2022-986-RC1)
**The reviewer also posted the same comments for the radiative manuscript (egusphere-2022-988).**

*The authors intend to investigate the influence of longwave radiation on evaporation. They claim that because longwave radiation is absorbed in the top 20 µm at the ocean surface this heat source/sink might cause significant deviations of the surface temperature from the underlying water. This effect changes the saturation water vapor pressure at the water surface. Therefore the concentration difference between the water surface and a reference height is influenced and with it the water vapor flux and evaporation rate. They claim that this potentially large effect is completely ignored in bulk formulas for evaporation.*

*The authors therefore want to study the effect of longwave radiation on evaporation in a wind tunnel specifically designed for this purpose. In the first paper (egusphere-2022-986) they focus on a thermodynamic characterization of the facility, and in a second one (egusphere-2022-986) on the radiative characterization.*

*The subject of the study is ill-defined and the described facility and instrumentation not really suitable for the intended purpose. Therefore the reviewer recommends rejection of the publication of the two manuscripts.*

**We thank the reviewer for the time taken to read the manuscript.**

**The review is clear-cut (e.g., ".. ill-defined ..", ".. not really suitable for the intended purpose ..", ".. rejection ..").**

**The very brief review does not include an assessment of any of the manuscript sections (2 Design and Operation, 3 Thermodynamic Evaluation, etc.).**

**Instead the review is based on an assertion that we are not aware of previous work. The (implied) implication is that the basic idea is fatally flawed.**

**The assertion/s are wrong.**

**Instead, we are conducting a series of very well-defined laboratory experiments as described below.**

*The claim that "mass transfer formulations for evaporation ... not directly consider the langwave radiative fluxes" is simply not correct. The reviewer did not perform a systematic literature search, but quickly found two almost 30 years old papers, dealing with the subject: Zhang and McPhaden 1995 (https://doi.org/10.1175/1520-0442(1995)008<0589:TRBSST>2.0.CO;2 ) and Fairall et al. 1996 (https://doi.org/10.1029/95JC03190). The actual version 3.6 of the COARE algorithm published on zenodo explicity includes longwave irradiation (named there IR flux): Bariteau et al., 2021 ([https://doi.org/10.5281/zenodo.5110991](https://doi.org/10.5281/zenodo.5110991))*

**The reviewer is incorrect.**

**To see that, let us examine the first work cited by the reviewer (Zhang and McPhaden 1995). In that work the calculation of the sensible (their Eqn 1a) and latent (their Eqn 1b) heat fluxes are reproduced from their paper as follows:**

$$F_h = C_p \rho C_h U (T_s - T) \qquad (1a)$$

$$F_q = L_v \rho C_e U [q^*(T_s) - q] \qquad (1b)$$

**With reference to (their) Eqn 1b, this is the classical Dalton-type bulk formula for evaporation that we referred to as the mass transfer formulation in the manuscript. Neither the incoming or outgoing longwave radiation is explicit in the equation. (Note: The cited Fairall et al 1996 and associated COARE reference cited by the reviewer use exactly the same mass transfer formulation (see Eqn**

17 in Fairall et al. 2003, J Climate 16: 571-591)). What the cited references do is calculate the evaporation using the mass transfer formulation (e.g., Eqn 1b above). They then combine the latent and sensible heat fluxes with the radiative fluxes to define the surface energy balance. We have done this ourselves on many occasions and this can be readily confirmed by a literature search. We note that this approach is described in standard texts (see references we cited on line 25 and line 40 of the manuscript) which present the same method as in the references cited by the reviewer.

However, we are going well beyond this long-used mass transfer formulation.

Instead, the fundamental basis for our new laboratory-based facility is described in lines 74-76 of the manuscript as follows:

> "The unique feature is an augmented capability to independently vary the incoming longwave radiation at the water surface whilst holding the other variables fixed. The scientific rationale of this approach was to isolate the effect of a change in the incoming longwave radiation on both evaporation and surface temperature."

We can explain this statement with reference to Eqn 1b (above). With $L_v$ (the latent heat of vaporisation) and $C_e$ (transfer coefficient) both more or less constant, one can envisage an experimental configuration where $\rho$, $U$, $q^*(T_s)$ and $q$ in Eqn 1b were all held fixed. Eqn 1b would predict no change in the latent heat flux if the incoming longwave radiation was independently varied because the incoming longwave radiation is not directly represented in Eqn 1b.

In our experiments we hold $\rho$ (air density), $U$ (wind), $q$ (specific humidity of air) and *T (temperature of air)* constant. We then vary the incoming longwave radiation and measure how the surface temperature ($T_s$) and latent heat flux respond. Hence our work is not "*ill-defined*". Instead we are conducting a series of well-defined laboratory experiments to experimentally examine the fundamental basis of the mass transfer approach itself. To our knowledge our work represents the first-ever experiment to examine the fundamental basis of the mass transfer approach. For this reason it is critical to fully document the technique/s - hence our submission to the AMT journal which seemed ideal for this purpose.

In response we have rewritten the introduction to make the approach even more explicit and have explicitly written the bulk-formula equation (new Eqn 1). We believe this should clear up any confusion.

*The authors are obviously not familiar with the extended research work on the difference between the ocean surface temperature and the underlying bulk water ("cool skin"). Much*

*of the pioneering work was done by Katsaros, see, e. g., Katsaros 1980 (https://doi.org/10.1007/BF00117914 ) or Katsaros 1990 (https://doi.org/10.1007/978-94-009-0627-3_9 ). A comprehensive account of the near-surface layer of the ocean is given in the monograph of Soloviev and Lukas 2014 ([https://doi.org/10.1007/978-94-007-7621-0](https://doi.org/10.1007/978-94-007-7621-0)).*

**In response we confirm that we are aware of the extended body of work (especially in remote sensing) on differences between the bulk and skin temperature over the ocean (and over lakes). However, again we re-iterate, we are examining a more fundamental question: the validity of the mass transfer formulation itself.**

*There are still, of course, open question. Most of them are related to the mechanisms of the transport from the ocean surface down to the bulk water, especially the influence of wind waves. The wind tunnel built by the authors is not suitable to address these questions because of the tiny and shallow water basin. A large wind-wave facility, such as the LASIF at the University of Marseilles (France) would be required for such studies (https://www.osupytheas.fr/?-LASIF-Grande-Soufflerie-air-eau-de-Luminy-&lang=en) and instrumentation and methods to image the water surface temperatures and temperature profiles in the aqueous viscous boundary layer.*

**We agree with the reviewer that there is much science still to be done on many important topics.**

**We also agree with the reviewer that the facility we have built is not suitable to address the influence of wind waves on heat transfer into the interior of the ocean. The reason is that this is not the scientific question we are addressing.**

**The question we address: the validity of the mass transfer formulation for evaporation that has been in widespread use for the last 220 years, is, in our opinion at least, a very important scientific topic. In that context, the experimental facility we have established represents the first-ever examination of the validity of the (220 year old) mass transfer formulation for evaporation.**

**Response to Reviewer Comment 2**
(https://doi.org/10.5194/egusphere-2022-986-RC2)

*Original review in ITALICS.*
**Author Response in bold.**

*A review of "Evaluation of a wind tunnel designed to investigate the response of evaporation to changes in the incoming longwave radiation at a water surface. I. Thermodynamic characteristics" by Michael L. Roderick, Chathuranga Jayarathne, Angus J. Rummery, and Callum J. Shakespeare.*

*In this manuscript, the authors describe an experimental setup for isolating the effect of longwave radiative flux on evaporation from a water surface. In short, the setup includes a wind tunnel containing a small water bath, with embedded sensors for measuring the humidity of the air and the temperature of the air and water. Additionally, incoming longwave radiative flux was measured via pyrgeometer and the water skin temperature was measured via microbolometer. The room containing the setup was described as being "temperature-controlled", though the cooling/ventilation system operated beyond the control of the authors, producing a noticeable oscillation in ambient temperature and humidity.*

*The topic as described by the authors is certainly of interest to the readership of AMT, and the manuscript was written with clear language. However, I have major concerns with some core elements of the laboratory setup. Furthermore, I found it difficult to assess the importance of this manuscript as an independent piece of research. The whole project is motivated by a desire to investigate the specific impact of longwave radiative flux on evaporation, but the details of the radiative component are left to the (as of yet unseen) part 2. I don't know that part 1 stands on its own as a meaningful contribution- is the result that bulk water temperature is sometimes close to the wet bulb temperature of the air above it? In any case, I believe that the work the authors have done can contribute the the body of knowledge, but I strongly recommend that they significantly revise this manuscript. Without seeing the second manuscript, I can only recommend that the revision to part 1 should include more details regarding the radiometric measurements (and results related to the total heat budget calculations). It may be that such a revision would combine the two parts- or that new radiative measurements need to be made with higher quality instrumentation, but it's impossible to say having only seen part 1 of the work.*

**We thank the reviewer for the careful and helpful review.**

**Both the second and third major comments made by the reviewer were good points that related directly to the radiative measurements that were the subject of a second (radiative) manuscript. We were able to adequately address the reviewers concerns using results taken directly from the second (radiative) manuscript.**

**In response we have combined both manuscripts into a single manuscript.**

*Major Concerns*

*The room that was described as "temperature-controlled" showed a regular oscillation of ambient air temperature and specific humidity (approximately 1000 second period). This has a meaningful impact on what should be a sensitive measurement. To this point, the oscillation in incoming longwave radiative flux appears to track (in both shape and phase) the oscillation in humidity- not temperature. Does this mean that radiative flux sensed by the pyrgeometer is representative of more volumetric (path) absorption/emission in conditions of elevated humidity? In any case- unless this periodic behavior can be leveraged as an asset in the heat flux balance calculations (i.e., the longwave radiative flux is the only oscillating flux, allowing the authors to parse its effect on the evaporation), I believe it will be a major liability in the calculations.*

**Close examination shows that the oscillation in longwave ($R_{i,F2}$ in Fig. 5e) due to the oscillation in the laboratory temperature control closely tracks the laboratory air temperature ($T_L$ in Fig. 5a, and equivalent to Black body at $T_L$ in Fig. 5e) and not the specific humidity. Hence the variations in specific humidity (e.g. $q_L$ in Fig. 5c) have no measurable impact on the longwave radiation.**

**The variations in specific humidity do follow the variations in laboratory air temperature but we know that the specific humidity has little impact on the radiative flux. We know that because at any given instant of time, the water vapour is more or less at the same temperature as the bulk air and the walls of the room (= $T_L$). Some of the (near blackbody) radiative flux emitted by the walls at $T_L$ would be absorbed by water vapour (also at $T_L$). The effective optical path length will change with the specific humidity as noted by the reviewer and the absorption will increase. But the absorption and re-emission by the water vapour occurs at the same temperature as the air and the walls of the room and this does not materially alter the longwave radiative flux. There is an in-depth explanation of the physics in the second radiative manuscript that is now included here as the moist air correction (Fig. 5). With that, we note that the temperature-controlled room behaves as a black body, all be it, with an oscillation induced by the heating system.**

**We handled the oscillation by taking the steady state period to be (approximate) integer multiple of the oscillation period. As a consequence the standard deviation for $T_L$ we reported (new Fig 12a) are overestimates. This is discussed in detail in the manuscript. We can extend that discussion here by using results from the original second radiative paper which note an overall uncertainty of around 2 to 3 W m$^{-2}$ in estimating the incoming and outgoing longwave fluxes at the water surface independent of the oscillation in the laboratory temperature. Hence any residual effect of the oscillatory behaviour**

**on the evaporation or longwave forcing is small relative the imposed longwave forcing at the water surface of 50 W m⁻².**

*Since the thin film-covered window only occupies a small segment of the hemisphere above the pool, how will the authors account for the difference between radiative flux emanating from the walls of the room and the flux emanating from the inside of the wind tunnel? Would a view factor (or some sort of solid angle accommodation) be helpful in accounting for this difference? Regardless, I'm not certain that a hemispheric pyrgeometer is the ideal sensor for this type of indoor, spatially heterogeneous measurement.*

**This is a perceptive comment. In response that is exactly what we did. We show below Fig. 10 from the second (radiative) paper which is now Fig. 8 in the combined paper.**

[Figure]

Figure 10: Schematic drawing showing separate contributions to the incoming longwave radiation at the water surface. The diagram is a cross section along the centreline of tunnel showing the hemispherical geometry used to estimate the incoming longwave radiation at the water surface arriving from the tunnel, film and PVC frame.

**To model the incoming longwave flux at the base of the tunnel we developed a theoretical model (see Fig above) that separately accounted for the longwave fluxes from the inside of the tunnel and from the room. The theoretical model was then evaluated using hemispherical measurements made using the radiometer located at the same position as the water bath. The theoretical model was found to predict the incoming longwave at the water surface with an RMSE of 2.2 W m⁻² (see new Fig. 9).**

*I mentioned the need for more information about the radiometry. It strikes me that measurements of the skin temperature are missing here. But even if they were included-*

*can the authors make a case that the FLIR E50 is up to the challenge of providing high-quality radiometric measurements? It appears to be a handheld system that is optimized for qualitative evaluation of heat sources in industrial/construction use cases. If the authors are able to provide the results of blackbody calibration to establish the instrument's accuracy, stability, and low noise, that might put these fears to rest. But non research-grade microbolometers are notorious for being poor in those categories and are exceptionally prone to drift (which might be the worst type of error one could have when making the "steady state dis-equilibrium" measurements described here). A cooled single-point infrared radiation thermometer is usually regarded as the superior instrument for these sensitive measurements.*

**The FLIR E-50 model has a least count of 0.05 degC. To independently test the accuracy, precision and overall repeatability we constructed a black body (a black painted copper plate) and by varying the temperature of the plate we could compare the flux measured with the FLIR E-50 with that measured by the Kipp and Zonen CNR1 net radiometer under a variety of conditions. We found the FLIR E-50 measurements to be repeatable and the recorded flux was the same (within error tolerances) as that of the research grade Kipp and Zonen radiometer.**

**As noted by the reviewer, the surface temperature measurements were missing here because they were originally included in the second radiative manuscript - they are now discussed in the new manuscript in several places (new sections 3.7, 4.3, 4.5).**

*Minor Comments*

- *For most parameters, variability is represented via standard deviation (or 95% confidence intervals). However, several quantities (ambient air temperature, ambient air humidity, incoming longwave radiative flux) oscillate with the room's cooling system. Perhaps the authors could report the amplitude of oscillation for these quantities?*

**Good point. For the variables mentioned (air temperature ($T_L$) and specific humidity ($q_L$) in the laboratory, incoming longwave flux at the top of the film ($R_{i,F2}$)) we have reported the standard deviation ($\sigma$) as the reviewer noted. The oscillation was the main source of (temporal) variation. With that, we note that the standard formula (for a sine wave, $Amplitude = \sqrt{2}\,\sigma$) can be used to estimate the amplitude.**

- *The variation in enthalpy (Figure 9) is computed via temperature difference between the beginning and end of steady-state period. What could be learned from*

*performing multiple short-window linear least-squares regressions during the steady state period, thereby obtaining a LHF estimate for each subwindow?*

**Good point. We tried this during our initial experiments and found the "noise" became too great over short time intervals in the LE (what the reviewer called LHF, latent heat flux) and also in the enthalpy change in the water bath (what we called *G*). Hence we were obliged to calculate the steady state values of the fluxes using longer periods that ranged from 850 (i.e., ~ 14 minutes) to 3300 (i.e., 55 minutes) seconds as described.**

*The Kipp & Zonen radiometer is said to be located 'in the laboratory (but outside the tunnel)' during evaporation experiments. Could this be pointed out in Figure 2?*

**During routine evaporation experiments the Kipp and Zonen radiometer was located inside the cardboard box sitting on the top of the wind tunnel (to the left of the number 3 in Fig. 2 as shown below). This was done to avoid thermal interference. For example, if the radiometer was sitting on top of the tunnel in the open, it would respond to the body heat of staff members as they walked past. Shielding the radiometer by placing it inside the cardboard box proved a simple solution that removed thermal interference. We have added a note to that effect in the new combined manuscript.**

[Figure]

90  Figure 2: Photograph of the wind tunnel in the temperature controlled room of the Geophysical Fluid Dynamics Laboratory. Key numbers as follows: [1] Water bath and digital balance (AND Corporation: Model GX-6100); [2] Variable speed fan; [3] Thermal camera (FLIR: Model E50); [4] Camera Spot (used for thermal camera calibration); [5] Radiator (for air temperature control); [6] Constant temperature water bath (Julabo: Model PP50); [7] Humidity/Temperature sensor (for measuring tunnel air, VAISALA: Model HMP140); [8] Humidity/Temperature sensor (for measuring laboratory air, VAISALA: Model HMP140); [9] Temperature
95  sensor (thermistor for measuring tunnel air, Thermometrics NTC: Model FP07DA103N ); [10] Vapour source (humidifier for humidity control of tunnel air); [11] Digital controller.

- *It is a bit taxing to jump between Figure 3 and the body of the manuscript to find the definitions for the state variables. I recommend adding descriptive labels on the figure or more content to the caption.*

**In the revised version we have also included a table of variables at the start (Table 1).**

- *For Figure 3, I think that a hashmark or dot pattern along the solid regions of the tunnel setup would aid the reader in interpreting the setup. Furthermore, arrows or streamlines in the tunnel portion would be helpful.*

**We have now combined the photograph with the schematic in a single figure (new Fig. 2) and we also have a Table of Variables (Table 1).**

- *Figure 11 is effective, but the extra information presented in Figure 12 is difficult to digest. Perhaps the authors could establish the concept in Figure 11 (as already done), then replacing Figure 12 with scatter plots that relate Tw, TB (or better, Tskin), and the inferred psychrometer constants.*

**This is a problem caused by having two manuscript parts that describe the (i) thermodynamics and (ii) radiation separately.**

**We have actually done what the reviewer suggested in the second radiative manuscript. We could not do that in the thermodynamic manuscript because the measurement of the surface "skin" temperature had not yet been described. We show Fig. 13 from the new combined manuscript below: with $T_B$ (x-axis) the mean bulk water temperature, and $T_S$ (y-axis) the measured 'skin' surface temperature.**

[Figure]

**Response to Reviewer 2 Comments on the radiative manuscript (egusphere-2022-988)**

*A review of "Evaluation of a wind tunnel designed to investigate the response of evaporation to changes in the incoming longwave radiation at a water surface. II. Radiative characteristics" by Michael L. Roderick, Chathuranga Jayarathne, Angus J. Rummery, and Callum J. Shakespeare.*

*In this manuscript, the authors continue their description of an experimental setup for isolating the effect of longwave radiative flux on evaporation from a water surface, focusing on the radiative characteristics of the setup. I apologize for the delay in posting my review; I needed to revisit the first manuscript and consider the two documents together.*

*In my review of the first manuscript, I raised a concern that the industrial handheld microbolometer (FLIR E50) might not be sufficiently sensitive or accurate to make the measurements the authors wish to make. Compounding this challenge is the presence of the window between the sensor and the sampling region. In order to isolate the testing volume from the ambient conditions in the laboratory while allowing for the longwave infrared camera to measure the water skin temperature, a window comprised of two ultra-thin polyethylene films was devised. To their credit, the authors recognize the need*

*to quantify the impact of this window on the radiative measurements, so a substantial portion of the present manuscript is devoted to quantifying this impact (vis-a-vis absorptive/reflective/transmissive properties of the window). In the end, the authors report a radiative heat flux uncertainty of ~3 W/m^2, corresponding to a skin temperature error of ~0.5 degrees Celsius. I find that result rather impressive, which seems to be a recurring theme in these two manuscripts: the authors invested a great deal of effort into setting up an experimental apparatus with less-than-ideal equipment in less-than-ideal ambient conditions, still managing to obtain good measurements through careful analysis and consideration of the sources of uncertainty. I believe there's value here, but find the clarity of the results to be muddled by the approach of presentation        (i.e.,        through        two        distinct        manuscripts).*

*Just as it was difficult for me to assess the impact or meaning of the first manuscript without knowing about the radiative measurements, it is difficult for me to place the results of the second manuscript into context without having the first manuscript open simultaneously alongside it. I don't think that's ideal, and I believe the readership of AMT would greatly benefit from a single, consolidated manuscript. The granular details (e.g., quantifying the impact of the film on the radiative measurements, calibration, demonstration of the process for computing uncertainty) shouldn't go to waste, but a good portion of the content could be relegated to an appendix in order to allow readers to have easy access to the material without it breaking up the flow of the paper. In any case- I'm neither an author nor an editor, so the decision to combine or not is left up to you. But it's difficult for me to imagine revisions to the separate manuscripts that would allow each to stand on its own.*

**We thank the reviewer for the helpful and supportive comments.**

**We have followed the suggestion by combining the two separate manuscripts into a single manuscript and we have made extensive use of appendices as suggested by the reviewer.**

**Michael L. Roderick & Callum J. Shakespeare (on behalf of all authors)**

**17th July 2023**

---

## Author Response (AR2)

**Manuscript Number: egusphere-2022-986**

**Title: Evaluation of a wind tunnel designed to investigate the response of evaporation to changes in the incoming longwave radiation at a water surface**

**Response to second round of reviews**

*Original review comment in ITALICS.*
**Author Response in bold.**

*Associate Editor (Dr Daniela Famulari)*

***Public justification (visible to the public if the article is accepted and published)****: I thank the authors for all the work they spent on synthesying two manuscripts into one, and for improving their initial contribution following the reviewers' inputs. I believe the work has improved considerably since the first submission, and the paper is in my opinion worth for publication, as it can provide useful discussion ground for this topic amongst the scientific community.*

**We thank the associate editor for the helpful and supportive comments. We agree it has been a long process but we also think the review process has improved the original submission/s and thank the reviewers for their contributions.**

*Reviewer 2*

*The authors have successfully integrated their two distinct manuscripts into a coherent single manuscript, the structure of which I find to be substantially improved to the point where I can evaluate it on its scientific merits. I won't rehash my description of the elements which existed in the two preceding manuscripts. However, I have some lingering "big picture" concerns:*

*The work is motivated by a desire to better describe the balance between evaporation and longwave irradiance at the air-water interface. In the discussion and conclusion sections of the manuscript, however, it remains unclear the specific research questions which are addressable via the setup described here. Zooming out even further- what is the impact of this work? I strongly recommend fleshing this out.*

*In support of those efforts, I make the following specific recommendations regarding the communication of quantitative results:*

*A fairly "low-cost" way of tying together the initial motivation and synthesis of results would be to plot curves associated with eq. 1 and then compute the size of the effect determined in your laboratory work due to radiative considerations.*

*Furthermore, I think the manuscript would greatly benefit from a comparison with the treatment of LW & SW radiative fluxes within the COARE parameterization. Whether or not the authors view COARE as the gold standard and the issue of parameterizing air-sea heat flux as settled, the inclusion of such a comparison will be of tremendous value to readers of this paper and help to place the results in a broader context.*

**We thank the reviewer for these helpful comments.**

**As noted by the reviewer below this is a "technique paper in a technique-focussed journal" and our aim was not to evaluate the science involved but rather the underlying techniques required before the big science questions could be addressed. The detailed analysis the reviewer refers to (i.e., an evaluation of the Dalton-type bulk formula) will appear in a future publication. Here we focus purely on the methods. Recall that we are not aware of any other study that has manipulated the longwave radiation**

independently of other variables (air temperature, humidity and wind). Our paper actually shows how difficult this was to achieve.

With that in mind we have modified the introduction by adding a further sentence to the justification paragraph (see new lines 66-67) where we have directly spelled out what this means. The new sentence we have added reads as follows:

"If the longwave fluxes were important for evaporation as we have inferred, but did not cancel, then the Dalton-type formulae in widespread use (e.g. Eqn 1) would not be a valid description of the evaporation process."

We believe that this very direct statement should satisfy the philosophy that underlies the reviewer comment/s.

*Minor stylistic/editorial comments:*

*Even though this is a technique paper in a technique-focused journal, I find that the overall course of the manuscript is bogged down by details which are ultimately not of central importance. For example, while the multiple depictions of radiative balances and corrections (Figures 4-7) exist to describe the considerations taken in designing the experimental setup, I believe that some of the figures corresponding to initial design iterations (e.g. single film) could be relegated to the appendices. This isn't a major sticking point, though.*

We respect the authors viewpoint and we accept there is a lot of material. We included the single film treatment (Figs 4, 5, 6) because that was used directly in the manuscript to experimentally estimate the transmission coefficient of the film. We felt that this important step had to be included in the main text but relegated the less important determination of the absorption/reflection to the appendices as originally suggested by reviewer 2.

*L52-54: it would be far more realistic (and not too onerous) to account for exponential decay of radiation: to first order, couldn't one assume that ~63% of the heat is absorbed between the interface and the e-folding depth? In a similar vein, I understand that this example exists solely to demonstrate the importance of evaporation, but the neglect of turbulence (even simple surface renewal) is striking. I recommend providing at least a rough order of magnitude estimate of the relative importance of the different heat transfer processes at play here.*

Respectfully we are unable to do this at the moment but it is something we plan for future publication/s. As explained in the introduction, the objective of the work is to evaluate the underlying validity of the Dalton-type formulae (Eqn 1). In the introduction we would be forced to use the Dalton-type formulae to calculate the relative importance of the various heat transfer processes involved.

*L623-624: parenthetical seems to be a relic from the two-manuscript organization*

Thank you. We have removed the parenthetical statement.

*Figure 15: I recommend that the ticks on the left and right axes are matched so that the gridlines are shared; an easy way to do this is to set limits of 1000-8000 Pa for e and 5-40 g/kg for q.*

We did not understand this point.

With the limits for $e$ (left hand axes) currently defined as 1000 to 6000 Pa, then the axes on the right (for $q$) cannot be arbitrarily assigned but is instead set automatically by the underlying physics linking $e$ and $q$. For example, at 1 bar total pressure the relation between $e$ and $q$ is:

| $e$ (Pa) | $q$ (g kg$^{-1}$) |
|---|---|
| 1000 | 6.24 |
| 2000 | 12.54 |
| 3000 | 18.87 |

[Figure]

```
4000    25.26
 …       …
```

**By changing the left hand axes as proposed the plot would look like:**

[Figure]

**We have left the original plot (Fig. 15) as is.**

**Michael L. Roderick & Callum J. Shakespeare (on behalf of all authors)**

**31st August 2023**

---

## Author Response (AR3)

**Manuscript Number: egusphere-2022-986**

**Title: Evaluation of a wind tunnel designed to investigate the response of evaporation to changes in the incoming longwave radiation at a water surface**

**Response to second round of reviews**

*Original review comment in ITALICS.*
**Author Response in bold.**

*Associate Editor (Dr Daniela Famulari)*

***Public justification (visible to the public if the article is accepted and published)**: Further small corrections are required following the recommendations provided by Reviewer #2.*

*Technical corrections:*
*I thank the authors for their rapid response, however I have to point out that, contrary to what was suggested, only a small part of what the reviewer#2 asked was satisfactorily addressed. In fact, only 2 small sentence modifications were made on the manuscript. I request that AT LEAST point 1 and 7 should be satisfactorily addressed before you resubmit your revised version. I understand that this has been a long process, but a last small effort to close the work would be preferable, especially considering the efforts both reviewers made to improve the manuscript.*
*Please, provide the following:*
*1-"In the discussion and conclusion sections of the manuscript, however, it remains unclear the specific research questions which are addressable via the setup described here. Zooming out even further- what is the impact of this work? I strongly recommend fleshing this out." Provide at least one sentence in the conclusions that clearly states what is the aim of the experimental device: currently you have provided a summary and synthesis of your setup, but which scientific question would you like to answer with this apparatus? What would it be useful for?*

**We have completely rewritten the final paragraph to directly address this comment. The new paragraph now reads:**

**In summary, the experimental system described here has been designed to investigate how evaporation is coupled to longwave radiation. In the traditional (Dalton-like) bulk formulae, evaporation is held to depend on the wind speed and the difference in specific humidity between the (near-saturated) surface and the ambient air. The traditional bulk formulae does not explicitly acknowledge any dependence on the longwave radiative fluxes. The experimental system can be used to hold the wind speed and specific humidity in the adjacent air at constant values while independently altering the incoming longwave radiation. By this design we are able to isolate any**

**direct coupling of evaporation to the longwave radiative fluxes. In the paper we have shown that the steady state wind tunnel system provides reliable measurements and we can impose a controlled longwave radiative forcing of around 49 W m$^{-2}$ that is known to within ±3.1 W m$^{-2}$. When combined with a measurement accuracy of the evaporative response to that forcing that will be better than 2 W m$^{-2}$ we conclude that the new wind tunnel system is suitable for the experimental investigation of the coupling of evaporation to longwave radiation.**

*2-3- You have responded to point 2 and 3, two possible suggestions on how to put into a wider context your work, however not introduced any change to the manuscript. The points raised would have been an improvement, but nothing strictly necessary to justify what you have done so far, you mentioned this is a methodology and experimental description, and the suggested points can also be addressed in a future contribution, therefore I accept your reply.*

**Noted.**

*4- You have not added what was asked, but not being a necessary point that is acceptable as well.*

**Noted.**

*5- You have not provided what was asked by the reviewer.*

**We have substantially modified the text (see lines 50-61 in the new track changes manuscript) to incorporate the idea of including the exponential dependence for the longwave absorption and by also noting that this applies even in perfectly still conditions.**

*6- OK, changed according to the suggestion.*

**Noted.**

*7- Not changed. The reviewer asked an aesthetic change to the chart in Fig 15, in order to make it more easily readable. The suggestion is to change the range of the y axis on the left (1000-6000) to 1000-8000), while leaving the axis on the right unchanged (range 5-40, as in the original chart). In the new chart provided in the reply, the right y axis range is also changed (5 to >50), defeating the purpose of the reviewer request. The point here is that the gridlines make it easy to be read on the left y axis, but not on the right: the*

*request is obviously NOT TO CHANGE ANY VALUES OF THE CURVE, but rather changing scales on the axes in order to make the plot more readable. Can you keep the range of right-y axis fixed, and change the left-y axis?*

**There is a misunderstanding here. The axes cannot be independently changed in the manner suggested since for a given total air pressure (= 1 bar), *e* (left hand axes) defines *q* (right hand axes) and vice versa. To help with the discussion below we note the following equivalent values:**

| *e* (Pa) | *q* (g kg$^{-1}$) |
|---|---|
| 801.4 | 5 |
| 1000 | 6.24 |
| 6000 | 38.19 |
| 6278.0 | 40 |
| 8000 | 51.31 |

**Now let us start with the original plot where the left hand axes has a range in *e* of 1000-6000 (Pa). On this choice the right hand axis range is automatically defined to be from 6.24 g kg$^{-1}$ (= 1000 Pa) to 38.19 g kg$^{-1}$ (= 6000 Pa). There is no choice.**

**Now assume we were to change the left hand axes range to be 1000-8000 (Pa) as originally suggested by reviewer 2. On this choice the right hand axes range will now be from 6.24 g kg$^{-1}$ (= 1000 Pa) to 51.31 g kg$^{-1}$ (= 8000 Pa).**

**Conversely if we set the right hand axes range to be 5 to 40 g kg$^{-1}$ then the left hand axes range is automatically defined to be 801.4 to 6278.0 Pa.**

**If we added grid lines for both sides the plot would like this:**

[Figure]

We personally find that having two sets of grid lines (one for each y axes) is a little more confusing than the original plot.

For that reason we have left the original plot but we can easily change it to the above if required.

Michael L. Roderick & Callum J. Shakespeare (on behalf of all authors)

10th September 2023